# Closing Kok's cycle of nature's water oxidation catalysis

Yu Guo[1,2], Lanlan He[1,2], Yunxuan Ding[1,2], Lars Kloo [3], Dimitrios A. Pantazis [4], Johannes Messinger [5,6] & Licheng Sun [1,2,7] ✉

The $Mn_4CaO_{5(6)}$ cluster in photosystem II catalyzes water splitting through the $S_i$ state cycle ($i = 0-4$). Molecular $O_2$ is formed and the natural catalyst is reset during the final $S_3 \rightarrow (S_4) \rightarrow S_0$ transition. Only recently experimental breakthroughs have emerged for this transition but without explicit information on the $S_0$-state reconstitution, thus the progression after $O_2$ release remains elusive. In this report, our molecular dynamics simulations combined with density functional calculations suggest a likely missing link for closing the cycle, i.e., restoring the first catalytic state. Specifically, the formation of closed-cubane intermediates with all hexa-coordinate Mn is observed, which would undergo proton release, water dissociation, and ligand transfer to produce the open-cubane structure of the $S_0$ state. Thereby, we theoretically identify the previously unknown structural isomerism in the $S_0$ state that acts as the origin of the proposed structural flexibility prevailing in the cycle, which may be functionally important for nature's water oxidation catalysis.

Biological water splitting in photosystem II (PSII) is catalyzed by the oxygen-evolving complex (OEC) that under illumination experiences four (meta)stable intermediates ($S_0$, $S_1$, $S_2$, $S_3$) plus one transient state ($S_4$), collectively known as "Kok's cycle". The OEC core comprises a $Mn_4CaO_5$ cluster in the dark-stable $S_1$ state, and the oxidizing equivalents are accumulated stepwise throughout the incremental S states until they are employed for water oxidation in the $S_3 \rightarrow (S_4) \rightarrow S_0$ transition[1,2]. As revealed by time-resolved serial femtosecond crystallography, in particular the X-ray free electron laser (XFEL) experiments[3–10], the $S_3$ state is structurally featured by an extra oxygenic ligand "Ox" on Mn1, which has been widely supposed to couple with the central μ-O5 for oxygen evolution in the following $S_4$ state[5,11–21]. After dioxygen release, the $S_0$ state is recovered by water insertion and proton expulsion, with the lowest metal oxidation states and unsaturated pentacoordinate Mn1 (Fig. 1).

For a long time, stories taking place during the crucial $S_3 \rightarrow (S_4) \rightarrow S_0$ transition had been rarely accessible to experimentalists. Very recently, remarkable experimental breakthroughs on this transition were reported by Bhowmick et al.[8] and Greife et al.[21], in which major clues for crystallographic structures and reaction kinetics have been provided. While "$S_3 \rightarrow S_4$" for generating the Mn(IV)−O• radical by single-electron, multi-proton transfer is identified as the kinetic bottleneck of "$S_3 \rightarrow S_0$" (ca. 2.5 ms), the subsequent $O_2$ formation and release during "$S_4 \rightarrow S_0$" are claimed to be much faster[21]. Although debates existed[22–26], the coupling between O5 and Ox (shown in Fig. 1) is regarded as the most viable mechanism for O−O bond formation in both reports. However, the cluster evolution after $O_2$ release remains unclear with respect to the possible intermediates involved and the atomic-level details of the transformations leading to reconstitution of the $S_0$ state, i.e., closing Kok's cycle. Consequently, the essential issues remaining in the

[1]Center of Artificial Photosynthesis for Solar Fuels and Department of Chemistry, School of Science, Westlake University, Hangzhou 310024, China. [2]Institute of Natural Sciences, Westlake Institute for Advanced Study, Hangzhou 310024, China. [3]Department of Chemistry, School of Engineering Sciences in Chemistry, Biotechnology and Health, KTH Royal Institute of Technology, SE-10044 Stockholm, Sweden. [4]Max-Planck-Institut für Kohlenforschung, Kaiser-Wilhelm-Platz 1, Mülheim an der Ruhr 45470, Germany. [5]Department of Plant Physiology, Umeå University, Linnaeus väg 6 (KBC huset), SE-90187 Umeå, Sweden. [6]Molecular Biomimetics, Department of Chemistry – Ångström Laboratory, Uppsala University, SE-75120 Uppsala, Sweden. [7]Division of Solar Energy Conversion and Catalysis at Westlake University, Zhejiang Baima Lake Laboratory Co., Ltd., Hangzhou 310000 Zhejiang, China. ✉e-mail: sunlicheng@westlake.edu.cn

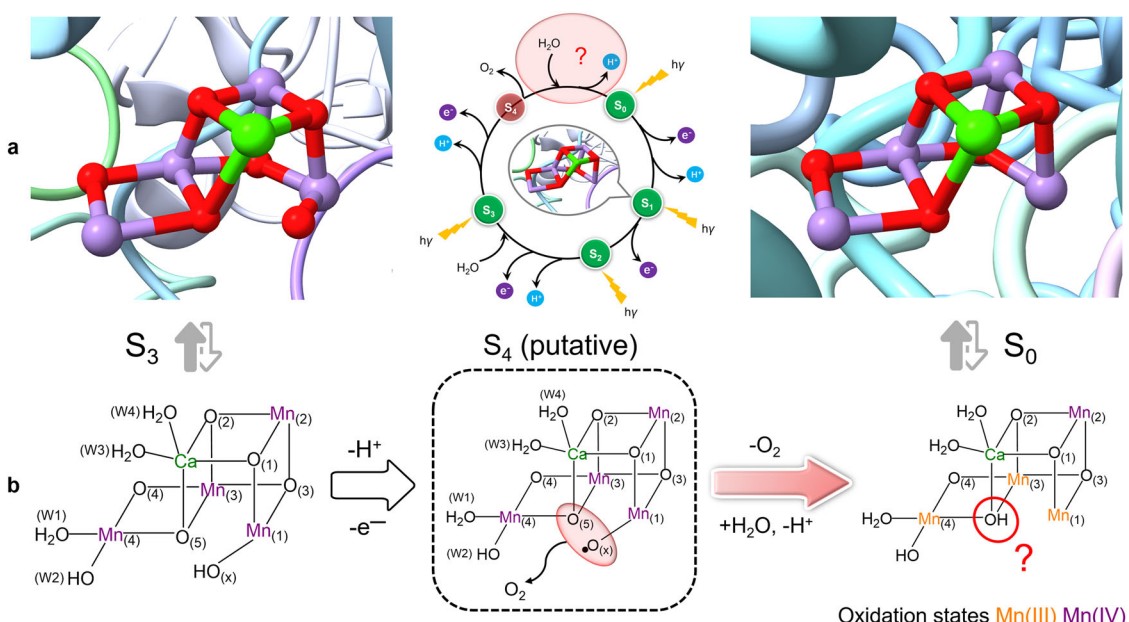

**Fig. 1 | Overview of the final transition of Kok's cycle. a** Exhibitions of the XFEL crystal structures of the OEC cluster in the $S_3$ and $S_0$ states[4,8]; the stage included in the $S_4 \rightarrow S_0$ transition after $O_2$ release (highlighted at the top of the cycle) represents the focus in this study. **b** Molecular diagram for the $S_3 \rightarrow S_0$ transition via a putative $S_4$ state where O–O bond formation occurs through O5–Ox• radical coupling (shown in the dashed box); the partial ligands in the $S_0$ state are not labeled because of possible ligand rearrangement during water insertion; the oxidation and protonation states are shown based on the most widely accepted forms[12,21,30,34,64,90].

recovery of the natural catalyst to its first catalytic state call for follow-up research.

Computational chemistry plays a prominent role in proposing realistic models for extremely fast processes that are difficult to be probed by experiments. Born–Oppenheimer ab initio molecular dynamics (BO-AIMD) simulations combined with minimum energy path (MEP) searches at the level of density functional theory (DFT) are employed in this work. These computations are logically based on the putative "oxo-oxyl coupling" mechanism in the $S_4$ state[11,21,27], at present the leading and most widely accepted proposal for $O_2$ formation (see Supplementary Note 1 for more discussion). Actually, the simulation results would be generally applicable in whatever coupling ways as long as O5 and Ox are substrates, which is most favored by the recent substrate water exchange experiments[28,29]. The main purpose is to investigate how the OEC cluster with a structural cavity left by release of $O_2$ (Im0$^{-O_2}$, Fig. 2) would evolve step by step, and thereby explore feasible pathway to the $S_0$ state. The study aims to shed light on the nature and origin of substrate water and provide insights on the underlying molecular mechanism resulting in the reconstruction of the Mn$_4$CaO$_5$ cluster, for a more complete understanding of nature's water oxidation catalysis.

To characterize the conformational changes of the cluster during water insertion in detail, our BO-AIMD simulations, without application of any steering force, execute a long simulation time based on a large model using full quantum mechanical treatment of DFT (Supplementary Fig. 1). BO-AIMD allows a priori exploration of the potential energy surface of a low-barrier transition through non-biased sampling, and hence identification of not already presumed reaction pathways and intermediates. The dynamic trajectories are strictly confined along the ground state under adiabatic approximation. It should be highlighted that the simulations cover only part of the "$S_4 \rightarrow S_0$" stage after $O_2$ release, instead of the "$S_3 \rightarrow S_0$" transition in a millisecond timescale that would not be captured in a picosecond trajectory (see more clarification in Supplementary Note 2). For further chemical reactions that may take place at timescales longer than tens of picoseconds of a typical AIMD simulation, MEP searches for specific reactions were carried out by truncated DFT models. The obtained results are beyond the current knowledge and discussed below.

## Results

### Water insertion dynamics and Mn$_4$Ca cluster evolution

Previous theoretical studies involving the $S_4 \rightarrow S_0$ transition have reached a general consensus that it is W3(H$_2$O), rather than W2 or any other crystal water molecule, that would refill the vacant site formed by $O_2$ release[30–33]. Since stepwise occurrence of $O_2$ release and water insertion is verified more favorable than the concerted mechanism[30,33], Im0$^{-O_2}$ is validated as the starting state pending water insertion (see more detailed discussion in Supplemenatry Note 4). Our BO-AIMD simulations for both octet/αααβ and doublet/αβαβ spin states (see Supplementary Note 5 for spin state definition) have observed interesting phenomena regarding the structural evolution of the cluster that have not been reported before. On that basis, water insertion can be divided into three basic stages, represented by Im0$^{-O_2}$ and the other two intermediates, Im1 and Im2 (Fig. 2). For the initial protonation state of W2, we here follow the hydroxide form W2(OH$^-$) that was suggested to be more consistent with the experimental magnetic and electron paramagnetic resonance (EPR) spectroscopic data[34,35] and the $pKa$ predictions by electrostatic energy computations[36]. However, it is noted that other studies that attempted to reproduce other types of spectroscopies such as Fourier transform infrared (FTIR)[37] and X-ray absorption spectroscopy (XAS)[38], and calculations of proton hyperfine coupling constants[39] favor the doubly protonated form W2(H$_2$O) in certain S states. Therefore, the nature of W2 remains an open question (see Supplementary Note 3 for more discussion).

Around the initial 5 ps of the simulations, the Ca-bound W3(H$_2$O) was found to insert into the structural cavity of Im0$^{-O_2}$, however, it would not directly migrate to the bridging position between Mn3 and Mn4, but moves towards Mn1 instead. Thereby, W3 forms a short, strong hydrogen-bonding (HB) interaction with W2(OH$^-$), while being in the bridging position between Ca and Mn1. This causes the square-pyramidal to trigonal-bipyramidal conversion of the Mn4(III) coordination, similar to the proposed five-coordinate Mn4(IV) in the local geometry[40,41]. Alongside, simultaneous movements of W5(HOH605) and W6(HOH577) were observed, occupying the original locations of W3 and W5 in close proximity, respectively (arrows in Im0$^{-O_2}$), and in this way, Im1 is formed. In the next 5–10 ps, the spontaneous proton

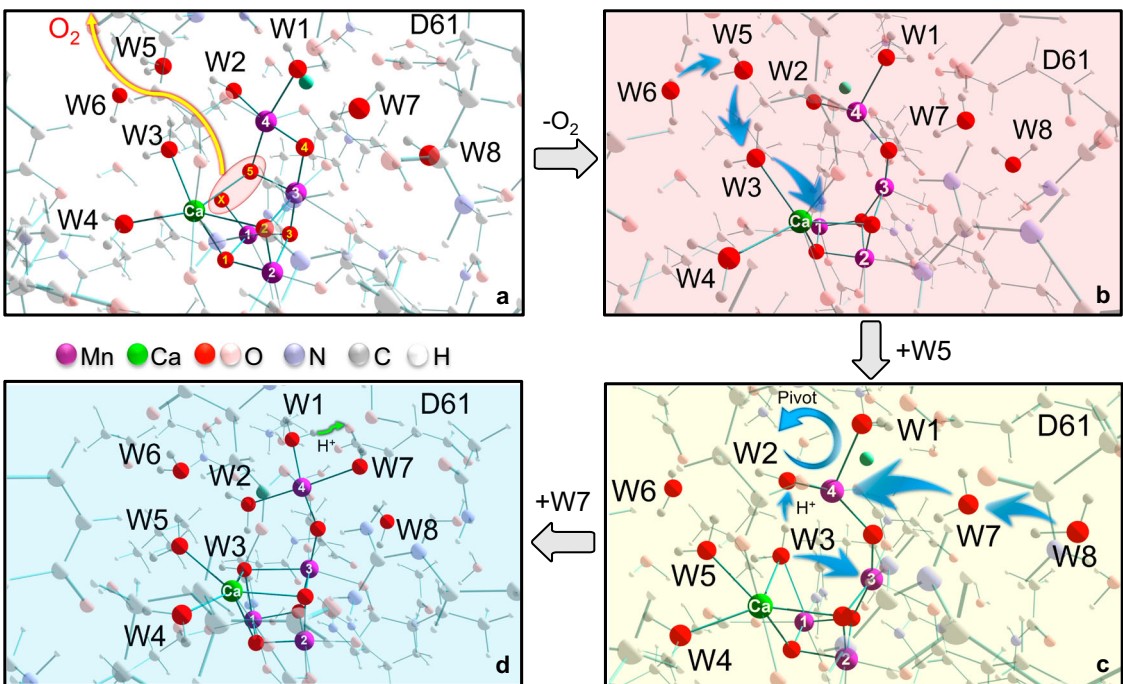

**Fig. 2 | Graphic presentations of water insertion after O₂ release. a** The S₄ state based on the XFEL structure of the S₃ state[6] upon removal of one proton and one electron on Ox; the yellow arrow represents the formation and release of O₂ through O5−Ox coupling. **b** The Im0$^{-O_2}$ state derived from the S₄ state by removal of O5 and Ox for O₂ release. **c** The Im1 state formed by W3 insertion to Mn1 and W5 binding to Ca. **d** The Im2 state formed by W3 deprotonation to W2 and binding to Mn3, and W7 binding to Mn4; the green arrow denotes H⁺ is suggested to be released to the lumen from W1 via Asp61. The routes of the key atomic motions observed during the simulations are marked with blue arrows. The red, yellow, and blue background colors stand for the three basic phases of Im0$^{-O_2}$, Im1 and Im2, respectively, which correspond to those in Fig. 3.

transfer from W3(H₂O) to W2(OH⁻) occurs, which was found to be highly correlated (almost synchronous) to W7(HOH523) binding to Mn4, and W8(HOH529) is also gradually pulled along that way (arrows in Im1). These structural changes are accompanied by a "pivot/carousel"-like reorganization[40,42,43] of the Mn4 ligands and expansion of the cluster with elongated Mn1-Mn4 distance (see Supplementary Note 6 for more analysis on the structural changes and energetics); at the same time, the deprotonated W3(OH⁻) approaches closer to Mn3 arriving at a bonding distance. As a result, Im2 is formed, which has a typical closed-cubane structure with a saturated octahedral Mn4 coordination and W3 as the Ca/Mn1/Mn3 μ₃-OH bridge group, and this conformation is dynamically stable thereafter. All these events were spontaneously take place during the simulations, indicating barrierless (or almost) events that can be easily captured in the dynamic sampling within tens of picoseconds (see Supplementary Movie 1).

The pathway observed for water insertion and structural transformation makes sense on the basis of molecular principles. The early formation of a closed-cubane (termed "B") instead of an open-cubane structure (termed "A"), i.e., that W3 initially binds to Mn1 rather than Mn4, can be expected for two reasons. Firstly, compared with Mn4, Mn1 is placed at a shorter spatial distance and with closer bond connectivity to Ca²⁺ (an indispensable cofactor for charge compensation in the cluster[44-47]), making Mn1 more positively charged than Mn4 (Supplementary Figs. 2 and 3) and thus a better Lewis acid for W3 coordination. Secondly, W2 was found to rotate moderately towards the cavity in the early phase of the simulations, which impedes W3 ligating to Mn4 but favors W3 binding to Mn1 by forming a strong HB interaction with it, as shown in Im1. Next, closing the Mn₃CaO₄ cubane is fulfilled by Mn3−W3 bonding, which necessitates W3 deprotonation (to W2) due to the stronger Lewis basicity of the deprotonated W3(OH⁻) bonding in the μ₃-position. Meanwhile, W3 movement to Mn3 further promotes the "pivot/carousel"-like rotation of W2 (in HB interaction with W3) together with W1 around the Mn4 axis, creating a

vacant site for W7 coordination from the O4 channel, which is a possible water delivery system[8,48-52]. Thus, it can be seen that the ligand rearrangements on Mn4 and the strongly coupled W7 binding with W2 protonation are attributed to the electrostatic affinity between Mn1/Mn3 and W3. Concurrently, these observations illustrate the critical roles of both Ca and Mn4 in water transport to the core position of the cluster. It is surmised that motions of outer crystal waters in HB network would also be involved, which is, however, not possible to be observed in a finite model (see Supplementary Note 7 for more discussion). Variations of the key interatomic distances with time evolution are displayed in Fig. 3. The electronic configuration of the cluster and metal oxidation states Mn1(III) Mn2(IV) Mn3(III) Mn4(III) remain unchanged throughout; see Mulliken spin populations in Supplementary Figs. 4 and 5. The almost indistinguishable phenomena for the two spin states reflect the insignificant role of the ferromagnetic/antiferromagnetic couplings between Mn2 and the other Mn centers. Furthermore, the commonality underlying the two simulations represents a mutual testification, eliminating the randomness and indicating the reproducibility of the results, and justifying the reliability of the conclusion.

With respect to the energetics of the three intermediates, additional geometry optimizations were applied to the snapshots extracted from the BO-AIMD simulations, showing an obvious downhill process as much as ca. −30 kcal mol⁻¹ in free energy difference from Im0$^{-O_2}$ to Im2 (Supplementary Table 1). However, it is inadequate to draw direct correlations in terms of the energetics to experimental measurements based on the present model, for which other parts and events during the S₄ → S₀ transition (e.g., O₂ release and H⁺ discharge leading to HB network rearrangement coupled to protein dynamics) beyond the local structural evolution surrounding the Mn cluster are not explicitly covered. Thus it is emphasized that the computed energetics do not fully represent the complete donor-side reactions of PSII[53], but only the intrinsic/local energetics of the conformational

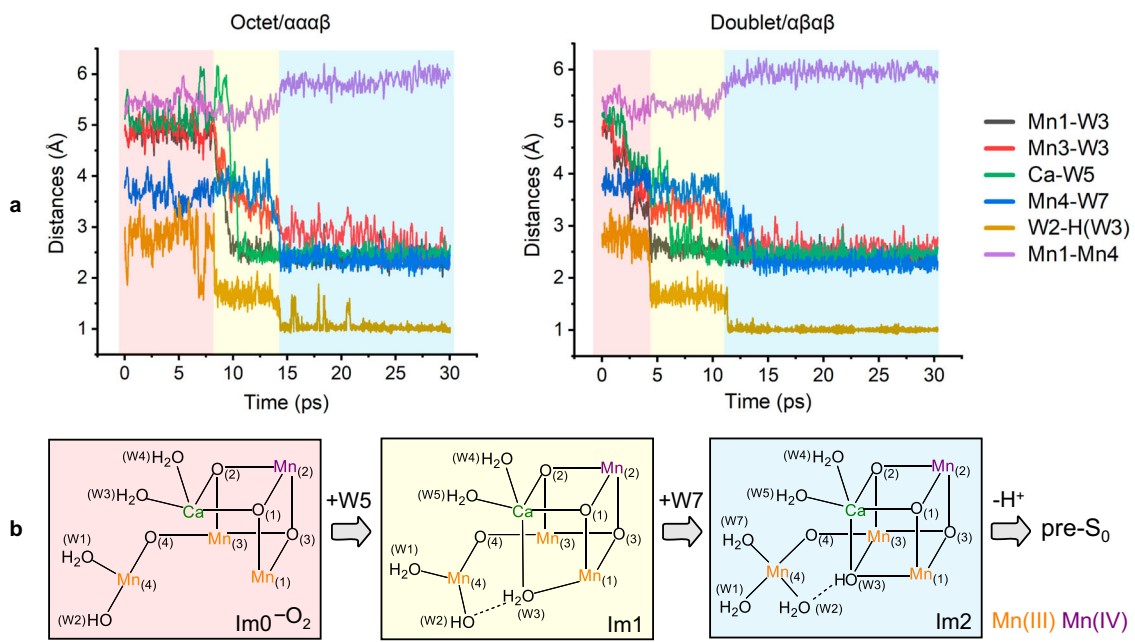

**Fig. 3 | Structural evolution of the Mn₄Ca cluster. a** Variations of the key interatomic distances with time evolution along the simulation trajectories for the octet/αααβ and doublet/αβαβ spin states. **b** Chemical structural formulas for the three intermediates; they all correspond to their respective local minima (stationary points) on the potential energy surfaces.

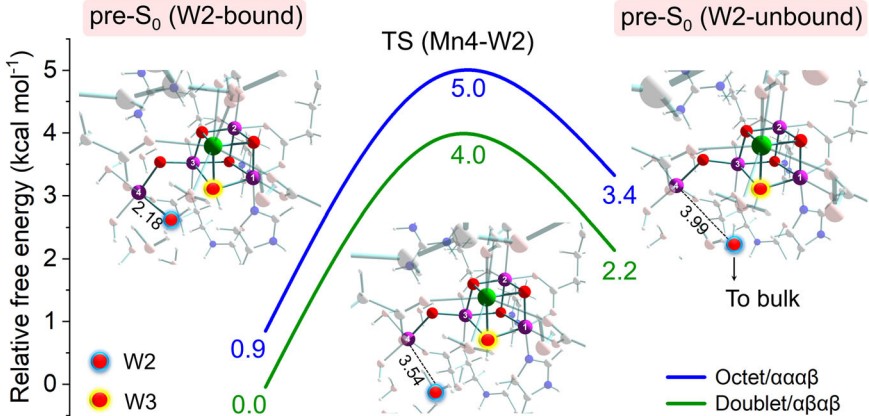

**Fig. 4 | Water (W2) dissociation in the pre-S₀ state.** Relative Gibbs free energy profiles for the octet/αααβ and doublet/αβαβ spin states are shown in blue and green, respectively; key interatomic distances are displayed in Ångström exemplified by the doublet/αβαβ spin state; these also apply for Fig. 5.

changes occurring at/around the inorganic cluster, and therefore the validity of the structural intermediates of the cluster from Im0$^{-O_2}$ to Im2 remain unaffected (see Supplementary Note 8 for more discussion). Anyway, the results demonstrate the thermodynamic rationality and kinetic feasibility for the ultrafast water insertion and cluster evolution after O₂ release. Thus, it is suggested here that W3 approaching Mn1 followed by formation of a closed-cubane structure, which is shown to occur at a picosecond timescale, is identified as the most favored pathway for water insertion under the present study. W1, now *trans* to μ-O4 in Im2, can easily deprotonate to the lumen via D1-Asp61 (arrows in Im2) seen from the obtained reaction energetics (Supplementary Note 9, Supplementary Table 2, Supplementary Figs. 8, 14 and 15). This is expected to be the second released H⁺ during the S₃ → S₀ transition, producing a state defined as "pre-S₀" herein, which will be studied further in the subsequent section.

**Attainability of the open-cubane S₀ state**

The above change of Im0$^{-O_2}$ → Im1 → Im2 is recognized as a very rapid conversion during the S₄ → S₀ transition. The pre-S₀ state formed by

the deprotonation of Im2 is still structurally different from the final S₀$^A$(open-cubane) state, because of its closed-cubane conformation and one additional water ligand. However, further progression from pre-S₀ to S₀$^A$, via stepwise W2(H₂O) dissociation and μ₃-W3(OH⁻) ligand transfer, are proven feasible (Fig. 4 and Supplementary Tables 3 and 4) by MEP calculations based on the evolved structures from the BO-AIMD simulations (Supplementary Figs. 9, 10, 16 and 19). As depicted in Fig. 4, W2 decoupling from Mn4 is estimated to be slightly endothermic by ca. 2–3 kcal mol⁻¹ with a small transition state (TS) barrier of ca. 4–5 kcal mol⁻¹, a magnitude that can be easily overcome by thermal vibrations in the protein matrix. According to indications from previous DFT studies in transition-metal chemistry[54–57], as well as our sensitivity test on functionals (see below), the calculations in this case can give quite reliable results with possible errors normally within a range of a few kcal mol⁻¹ that would not affect the feasibility of the reaction. Ligand dissociations are ubiquitous in organometallic chemistry, mostly acting as pre-steps in substitution reactions for formation of catalytically active species as necessary intermediates[58]. The obtained barrier height for W2 detaching from Mn4(III) is quite

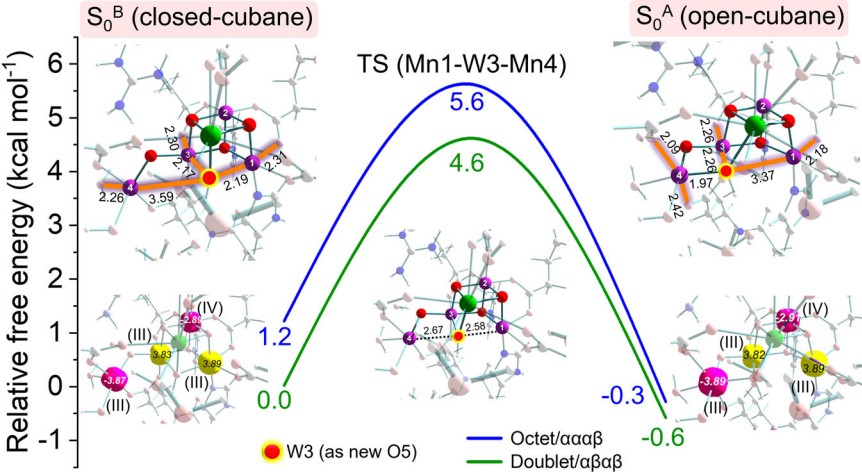

**Fig. 5 | Structural isomerism by μ₃-OH⁻(W3) ligand transfer in the S₀ state.** Metal oxidation states are labeled in Roman numerals and the J–T elongation axes are marked in orange. Spin up/down is marked in yellow/burgundy at the atomic centers involved (contour value for spin density set to 0.1). Note that all these parameters show slight model/functional dependence (see below).

similar to a recent report on a biomimetic polyoxometalate water oxidation catalyst[59], and the reactivity is enabled by the presence of Jahn–Teller (J–T) effect at Mn(III) which extends the bonding distance of the J–T axial ligand and thereby facilitates its de-coordination. Besides, the coincidence of W2 protonation and W7 binding synergistically further weakens the W2 coordination to Mn4, because of the further elongated Mn4–W2 bond and the "structural *trans* effect"[60,61] in octahedral transition-metal complexes. Formation of the pre-S₀ (W2-unbound) state, with W2 moving out to a distance about 4 Å away from Mn4, is analogous to the last necessary step required in the substrate water exchange mechanism where a water molecule dissociates from Mn4(III) in a closed-cubane structure (either exoergic (S₁) or endoergic (S₂))[62]. Furthermore, the possibility of W2 decoupling from Mn4(III) in a closed-cubane structure is also reflected in a hypothetical mechanism for water exchange in the S₀ state[63]. Thus we consider the dissociation of W2(H₂O) at this state as a chemically sensible process (see Supplementary Note 10 for more discussion). In this way, W2 is released to the bulk and the S₀ᴮ(closed-cubane) state is formed.

Thereby, Mn4(III) in the resulting S₀ᴮ structure contains an unsaturated peta-coordination sphere which allows for the interaction with μ₃-W3(OH⁻). As shown in Fig. 5, the μ₃-W3(OH⁻) shift from Mn1 to Mn4 can be realized across a low barrier of ca. 4−6 kcal mol⁻¹, reaching the almost isoenergetic S₀ᴬ structure that is perceived as the generally accepted S₀ state structure[4,30,34,64], and the intermetallic distances (2.77, 2.76, 2.84 and 3.40 Å) are basically consistent with the structural constraints from the extended X-ray absorption fine structure (EXAFS) experiments[65,66]. According to the Eyring–Polanyi equation, the reactions should occur at a timescale of nanoseconds at room temperature, which is within (far below) the experimental limitation of ca. 2.5 ms observed for the S₃ → S₀ transition[21]. Judging from thermodynamics and kinetics, the isomerization is readily reversible, switching the orientation of the J–T elongation axis of Mn4(III) along the W7−W3 or Asp170-Glu333 vector, while retaining those of Mn1(III) and Mn3(III) perpendicular to each other. Therefore, the conformational interconversion in S₀, with regard to mechanism, resembles that of the S₁ state induced by the J–T effect[67] and fundamentally differs from that of the S₂ state where valence isomerism (III) ↔ (IV) between Mn1 and Mn4 is involved[68].

Moreover, the above findings have been exposed to a sensitivity measurement where the outcomes by different DFT functionals have been examined using a series of different dispersion-parameterized hybrid and nonhybrid functionals (Supplementary Note 11, Supplementary Table 5 and Supplementary Fig. 11). In addition, a similar test has been performed to verify the structural isomerism using another structural model originating from the 3 F XFEL data (PDB ID: 6DHP)[4] with the S₀ state as the major population (Supplementary Table 6 and Supplementary Figs. 12 and 13). The obtained thermodynamic parameters are generally in accordance with that of Fig. 3, even though slight variations depending on model/functional are as expected observed (deviations within 1−2 kcal mol⁻¹), and such minor deviations would normally be expected given the inherent approximations and limitations of the DFT methodology[54−57]. Although it is not possible to offer an absolute quantification of the reaction energetics, a reliable qualitative conclusion can be safely drawn, i.e., W2 dissociation is achievable, and S₀ᴬ and S₀ᴮ are quasi-isoenergetic and interconvertible through a low barrier. Thus we emphasize the structural isomerism as a basic function of the OEC that is already manifested in the first state of the catalytic cycle.

## Discussion

Bhowmick et al.[8] and Greife et al.[21] have unveiled key structural and kinetic data during the millisecond S₃ → S₀ transition. According to Bhowmick et al., from 1200 to 4000 μs several structural changes indicative of O₂ release and/or water insertion occur, and the extended timescale between 2000 and 4000 μs may be due to pronounced variation in water positions (e.g., the slow reappearance of W20 in the O4 channel) and significant rearrangement of the HB network (including water-water, protein-protein, and water-protein interactions) related to the last proton release. This precisely corresponds to the suggested H⁺ transfer from Im2 to the lumen *via* D1-Asp61 in our scheme, but the further effect on the broad water/protein environment in PSII cannot be embodied in the present study. This may provide clues for capturing more possible intermediates during the extended timescale for crystallographic snapshot data in future. As no structural evidence was observed for an empty O5 site, it is speculated that refilling of the cavity by water binding is ultrafast, which is also reflected in our simulation results. Within this period, 1200 and 2000 μs are the two essential timepoints that are closely related to (but not entirely covered by) our work. The 1200 μs snapshot signifies the onset of O₂ evolution, and the 2000 μs snapshot, without Ox on the electron density omit map, indicates completion of binding of a water molecule that refills the vacant site formed by O₂ release. On this basis, the series of processes we suggest in the present study should in principle transiently reside between the two timepoints, but represent a very short phase seen from the picosecond water insertion and

subsequent nanosecond isomerization of the cluster (see Supplementary Note 16 for more discussion). This complements the details en route to the $S_0$ state, although the proposed temporarily present species may not accumulate to sufficiently large amounts to be detected experimentally by XFEL crystallography with microsecond intervals for snapshots. Since we show above that the $S_0^B \to S_0^A$ closed-to-open conversion in nanoseconds appears as the kinetically most demanding step after $O_2$ release, the whole cluster transformation from $Im0^{-O_2}$ to $S_0^A$ also satisfies the requirement that the rate-limiting step of $S_3 \to S_0$ in millisecond timescale belongs to the $S_3'Y_Z^+$ to $S_4$ transition prior to O−O bond formation (rather than $S_4 \to S_0$), as determined by Greife et al. As the specific locations of water molecules and HB interactions play a very important role for facilitating the critical step of Mn(IV)-O• formation during the $S_3 \to S_4$ transition, they are also crucial for the resetting process of the $Mn_4CaO_5$ cluster after $O_2$ release, especially for the W3-W2 interaction partially accounting for W3 binding to Mn1 and the internal proton transfer further promoting the Mn4 ligand reorganization (Supplementary Note 6). Besides, the HB interaction of W3−W5, of W5−W6, and W7−W8 (and similarly others outside the present model) are also indispensable for recovery of the microenvironment of the OEC cluster. Different from the single-electron multi-proton transfer with a moderate energetic barrier (13.6 kcal mol$^{-1}$) for the $S_3'Y_Z^+$ to $S_4$ transition, the present case does not involve any apparent electron transfer (no electron-hole to be reduced) and involves only one barrierless (or almost) proton transfer from $W3(H_2O)$ to $W2(OH^-)$ on the picosecond timescale, which appears much less demanding than the deprotonation of Ox−H. This is also consistent with the fast kinetics for the $S_4 \to S_0$ transition post to O−O bond formation (see Supplementary Note 17 for more discussion).

Previous to the present study, Capone et al.[33] have conducted a molecular dynamic study on the mechanism of oxygen evolution and $Mn_4CaO_5$ cluster restoration, based on the oxo-oxyl coupling mechanism for O−O bond formation. The major conclusions include confirmation of the Ca-bound W3 as the inserted water molecule and validation of the two-step mechanism for $O_2$ release and water insertion, which are both in line with this study. For restoration of the $Mn_4CaO_5$ cluster, they assume that W3 (along with deprotonation to W2) moves to the bridge position of Mn3/Mn4, directly resulting in the open-cubane structure of the $S_0$ state, which coincides with the schemes by refs. 30−32 in terms of the binding sites of W3 throughout. However, here we propose a different pathway for how W3 should enter the cavity, i.e., it binds first to Mn1 for formation of the closed-cubane intermediates followed by W2 protonation, which triggers the ligand reorganization on Mn4 and the coordination of W7, and ultimately the open-cubane $S_0$ state forms after $W2(H_2O)$ dissociation and then $W3(OH^-)$ transfer to Mn4. While uncertainty remains regarding the cause of the differences, the two plausible pathways should be noteworthy currently and further comparative studies may be needed. Anyway, since the target structure is the same, the alternative pathway presented here reveals possible intermediates and the important structural isomerism for the first state of the catalytic cycle (see Supplementary Note 18 for more discussion).

It is noted that the XFEL crystallography resolved only open-cubane structures[4,8,69] and only multiline EPR signal is exhibited[64,70−75] for the metastable $S_0$ state. Besides, the $^{55}$Mn hyperfine, nuclear quadrupole interaction, and EXAFS parameters are best matched by those estimated for the $S_0^A$ conformation[65,66]. However, these do not conflict with the presence of the structural isomerism. Assuming that decontamination is sufficiently complete and the resolution is sufficiently high for the targeted $S_0$ state, one straightforward interpretation would be that $S_0^A$ dominates over $S_0^B$ in the XFEL samples. According to the relationship between $\Delta G°$ and equilibrium constant $K$, a very minor energy advantage of one isomer over the other (that is even within the DFT error range) would lead to its overwhelming proportion; see Supplementary Note 11 for an analysis on relative populations and estimated energetics. Furthermore, Cox et al. pointed out that it is possible that a certain state that can exist in multiple conformers may not show all these forms under the experimental conditions[76]. Ibrahim et al. stated that the implicit form (if present) may be short-lived due to fast formation and decay kinetics, and/or its fraction may be below the detection limit[6]. Actually, the theoretically proposed $S_0$ structural isomerism is experimentally suggestive in several aspects. The $S_0$ multiline signal in spinach is only visible in presence of a few percent methanol[70−75], which is a strong indication for an equilibrium between at least two states that are close in energy. The suggestion is also supported by the fact that in thermophilic cyanobacterium *T. vestitus* the $S_0$ multiline signal can be observed also in absence of methanol[74]. Besides, the closed-cubane $S_0^B$ conformation is entailed in elucidating the water exchange mechanism[63], similar to that of the other S states[16,28,62,77]. Thus the presence of the $S_0$ isomerism makes sense even without a crystallographic structure available for a closed-cubane form. The situation is reminiscent of a recent experimental (EPR) evidence presented by Kosaki and Mino[78] (in line with Saito et al.[79] but in contrast to Barchenko and O'Malley[80]) identifying the $g = 4.1$ $S = 5/2$ high-spin $S_2$ state as a strong support for closed-cubane conformation, which is, though, also still unidentified in the XFEL studies.

Although not yet resolved by XFEL crystallography[3−8,81,82], closed-cubane structures could silently play a vital role in the catalytic cycle; see Supplementary Note 13 for additional discussion on the validity of this hypothesis. As illustrated in Fig. 6, our theoretical simulations suggest a likely missing link during the $S_4 \to S_0$ transition in which closed-cubane intermediates are involved, and furthermore the isomerism in the $S_0$ state, in addition to those already well established for the $S_1$ and $S_2$ states[67,68]. Hereby, the possibly final knowledge gap of the structural flexibility exhibited by the $Mn_4CaO_{5(6)}$ cluster is filled, thus serving as the open-closed structural precursors for the following S-states. Since the transformation is reversible, the isomerism provides a basic reference for a mechanistic proposal of O−O bond formation in either open- or closed-cubane structure, if they are both catalytically relevant. It is inferred that different from most artificial compounds, the exclusive ability of geometric rearrangement should be a unique characteristic of the tetranuclear manganese OEC cluster, and may be associated with its inherent excellent catalytic efficiency (the correlation to be further studied). The theoretical discovery of such potential structural variants can motivate new designs and improvement of bio-inspired water-splitting catalysts.

In brief, we have theoretically explored the chemistry in the $S_4 \to S_0$ transition after $O_2$ release and provided important information for the $S_0$-state reconstitution. The identified pathway for the insertion of the Ca-bound water molecule W3 into the cavity of the OEC cluster involves closed-cubane intermediates and is energetically favorable. This is accompanied with spontaneous $W3(H_2O)$ deprotonation to $W2(OH^-)$ and a series of water molecule displacements. The subsequent $W2(H_2O)$ dissociation and $W3(OH^-)$ shift to Mn4 are facile, making the open-cubane $S_0$ state for the next cycle attainable. More importantly, the results from this study encourage us to propose the reversible isomerization in the $S_0$ state that is concomitant with alteration of the J−T distorted axis of the dangler Mn4(III). The isomerism, already existing in the first state of Kok's cycle, lays the structural foundation for the subsequent S-states and may contribute to water oxidation catalysis in PSII.

## Methods
### BO-AIMD simulations
The BO-AIMD model for water insertion after $O_2$ release (depletion of O5 and Ox for the $O_2$-released state) are based on the room-temperature serial femtosecond crystallography of the $S_3$ state, which was taken from the second flash (200 ms) data by Ibrahim et al.[6]

**Fig. 6 | Suggested missing link and isomerism in Kok's cycle.** The missing link during the $S_4 \rightarrow S_0$ transition embedded in the full catalytic cycle is highlighted in red, and the isomerism in green. O5 in $S_1^B$ is at a non-bonding distance with respect to both Mn1 and Mn4[67]. '⇌' in the $S_3$ state denotes the unidirectional iso-merization because the open-cubane structure is largely stabilized[12,91,92]; however, our recent study suggests the isomerization becomes reversible again in the ensuing $S_3Y_Z\bullet$ state after H+ release[93]. O−O bond formation in the $S_4$ state (peroxo in

$S_4'$) for either open-[11,94–97] or closed-cubane[27,31,98–100] structure (with O5 and Ox as substrates) has been theoretically supported, and the Im0$^{-O_2}$ structure formed after $O_2$ release would be basically the same; thus the missing link would apply in either case. Note that the protonation states for some certain ligands are still debatable, such as W2 = OH−/$H_2O$[34–38,101,102], O4/O5 = $O^{2-}$/OH− (for $S_0$)[34,37,38,64,103], and Ox = OH−/$O^{2-}$/O− (for $S_3$/$S_3Y_Z$')[4,5,12,38,97,104], etc.

(PDB ID: 6W1V, monomer A) after removal of the mixed $S_2$ state in minor population. The model consists of the inorganic $Mn_4CaO_4$ cluster, 20 amino acid residues Asp61, Asn87, Tyr161, Gln165, Ser169, Asp170, Asn181, Val185, Glu189, His190, Asn298, Lys317, His332, Glu333, Ala336, His337, Asp342, Ala344, Glu354, and Arg357, and 24 crystal water molecules HOH515(W1), HOH530(W2), HOH584(W3), HOH550(W4), HOH605(W5), HOH577(W6), HOH523(W7), HOH529(W8), HOH525, HOH596, HOH511, HOH626, HOH627, HOH522, HOH534, HOH600, HOH514, HOH517, HOH505, HOH574, HOH541, HOH508, HOH526, HOH545 and one chloride ion (Cl−407), resulting in total 369 atoms and a net total charge of +1 (Supplementary Fig. 1). The quantum mechanical (QM) size is by far the largest one among all molecular dynamic studies on the OEC, and the specially customed GPU acceleration enables a long simulation time on such a large QM model. Since the focus of the computational study is within or closely around the OEC cluster, the large model in full QM treatment is capable of representing the conformational changes of the cluster affected by water insertion. Protonation states were chosen according to the most widely accepted scheme[18,21,34,62,83]. The $O_2$-depleted model

of the $S_4$ state, which is derived from the XFEL structure of the $S_3$ state by removing H+ and e− from Ox, renders the initial coordinates for the Im0$^{-O_2}$ intermediate. The models for the energetic comparisons among Im0$^{-O_2}$, Im1 and Im2 were extracted from the starting structure of the first phase, the last snapshot for the second phase, and the last snapshot for the third phase shown in Fig. 3, which were then fully optimized in the same size as used in the BO-AIMD simulations. The dispersion-corrected density functional theory, DFT-D3, using the hybrid functional B3LYP (in its standard form) was performed at double precision with the hybrid DIIS/A-DIIS scheme. The double-$\zeta$ effective core potential (ECP) basis set LanL2DZ was applied for the metal atoms Mn, Ca and the mixed full electron basis set 6−31 G*/3−21 G was applied for the H, C, N, O, Cl atoms (3−21 G is only used for the alkyl groups of the peripheral residues that are non-bonding to the cluster). Energy minimizations using the limited-memory Broyden−Fletcher−Goldfarb−Shanno (L-BFGS) method were implemented before the BO-AIMD simulations at the same level of theory. Backbone constraints by fixing α-carbons of peptide bonds were applied throughout the computations. The model is placed in a

spherical cavity surrounded by the conductor-like screening model (COSMO) of the polarizable continuum with a dielectric constant $\varepsilon = 6.0$, implicitly mimicking the protein matrix. The Bussi–Parrinello Langevin dynamics ($T_{damp} = 1000$ fs) for the thermostat and time-reversible integrator with dissipation for self-consistent field (SCF) were adopted for the NVT canonical ensemble (particle Number, Volume, and Temperature) simulations at 298.15 K. The simulations were pursued for at least 30 ps with velocity-Verlet integration using a 1.0 fs time step. All the BO-AIMD simulations were executed by the commercially available GPU-accelerated package TeraChem[84] (version 1.94) on supercomputers equipped with NVIDIA Tesla V100 cards.

## MEP calculations

Truncated DFT models were constructed from the last snapshots of the BO-AIMD simulations for the subsequent MEP searches, without missing out possible influence from surrounding necessary groups. This approximation represents a sensible compromise based on consideration on the feasibility and efficiency for massive Hessian calculations. For W1 (the new W2) deprotonation to Asp61, the model includes the inorganic $Mn_4CaO_6$ cluster, 10 amino acid residues Tyr161, Glu189, His190, His332, Glu333, His337, Asp342, Ala344, Glu354, and Arg357, Asp61 and 12 crystal water molecules W1-W8, HOH596, HOH545, HOH525, and HOH526, resulting in total 223 atoms and a net total charge of +1 (Supplementary Fig. 8). According to the test by Retegan et al.[85], a QM model with such a size is adequate to yield accurate reaction energetics and spectroscopic properties for the OEC system. For the subsequent $W2(H_2O)$ dissociation from Mn4, the starting geometry for MEP was based on the product state of Asp61 protonation by exclusion of the protonated Asp61 from the model (the reason is described in Supplementary Note 14, Supplementary Figs. 9 and 10). For the later $\mu_3$-$W3(OH^-)$ ligand transfer for the closed-open-cubane transition, the starting geometry for MEP was based on the product state of W2 dissociation by further removal of W2 (assumed to release to the bulk). Test computations based on the original crystal structure 6DHP including Asp61 in the model justify the validity of the conclusion (Supplementary Figs. 12 and 13). The proton-released state is termed "pre-$S_0$", as shown in Fig. 3, specifically pre-$S_0$ (W2-bound) and pre-$S_0$ (W2-unbound) according to the bound state of W2 in Fig. 4. In the next step, the $S_0^B$(closed-cubane) state is formed after W2 is released to the bulk, and then the $S_0^A$(open-cubane) state is formed by $\mu_3$-$W3(OH^-)$ ligand transfer. Geometry optimizations with backbone constraints ($\alpha$-carbons fixed along the peptide chains) were performed by the unrestricted hybrid functional B3LYP*[86] (15% exact exchange, dispersion parameters taken from B3LYP, Supplementary Note 11), supplemented by various different dispersion-parameterized DFT functionals for comparison where the obtained relative free energies are small (within a few kcal mol$^{-1}$). The LanL2DZ and 6–31 G* basis sets were used for Mn/Ca, and the rest H, C, N, O, and Cl atoms, respectively. Analytic frequency calculations on the optimized structures at the same level of theory verified all local minima; zero-point energies (ZPE) and thermal effects (298.15 K, 1 atm) were also extracted for thermal correction to Gibbs free energies. Transition states were confirmed through eigenvectors with adequate and expected negative eigenvalues, single imaginary frequency vibrations, and intrinsic reaction coordinate (IRC) analyses (see Supplementary Note 15 for more details), ensuring that the relationship with reactants and products are logical and correct. TSs were located by the Berny algorithm and synchronous transit-guided quasi-Newton (STQN) method. Finally, more accurate single point energies were computed with SDD (for Mn/Ca) and cc-pvtz (-f) (for H, C, N, O) basis sets under SMD continuum solvation model (solvent-accessible surface, $\varepsilon = 6.0$). DFT-D3 with Becke–Johnson (GD3BJ) damping dispersion correction was applied in both geometry optimizations and single-point energy calculations. All these computations were carried out by Gaussian 16 (version C. 01)[87]. Formal oxidation states for Mn were verified by Mulliken spin populations and localized orbital bonding analysis (LOBA)[88] using Multiwfn[89] (version 3.8).

## Reporting summary

Further information on research design is available in the Nature Portfolio Reporting Summary linked to this article.

## Data availability

All data generated in this study are available in the Supplementary Information or from the corresponding author upon request. Source data containing key trajectory information during BO-AIMD simulations and the Cartesian coordinates of optimized structures in MEP calculations are provided. Source data are provided with this paper.

## Code availability

The codes used for ab initio MD simulations (TeraChem[84], version 1.94, commercially available, www.petachem.com), geometry optimizations and MEP calculations (Gaussian[87], version 16 C.01, commercially available, www.gaussian.com), and LOBA (Multiwfn[89], version 3.8, open source, sobereva.com/multiwfn) are described in the Methods, and the Fortran code (customized) used to capture the interatomic distances along the dynamic trajectories is provided in the Supplementary Information.

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

## Acknowledgements
This work is financially supported by the National Key Research and Development (R&D) Program of China (2022YFA0911900), the Research Center for Industries of the Future (RCIF) at Westlake University, and the starting-up package of Westlake University. We thank Westlake University HPC Center for computation support. D.A.P. acknowledges support by the Max Planck Society.

## Author contributions
L.S. designated and supervised the study. Y.G. performed all the computational modeling and wrote the manuscript. L.H., Y.D., and L.K. collected and analyzed the BO-AIMD data. D.A.P. analyzed the MEP data and validity with respect to the structural isomerism. J.M. analyzed the significance of the findings and rationalized the isomerism in relation to experimental observations. All the authors contributed to the revision of the manuscript.

## Competing interests
The authors declare no competing interests.
