## [Peer Review File · Nature Communications]

Closing Kok's cycle of nature's water oxidation catalysisREVIEWER COMMENTS

Reviewer #1 (Remarks to the Author):

Review on "Closing Kok's Cycle of Nature's Water Oxidation Catalysis" by Guo et al.

The process of light-driven water oxidation by photosynthetic organisms is crucial to life on Earth; its byproduct dioxygen has shaped the atmosphere and geosphere. Recent advances in experimental and theoretical work on photosynthetic water oxidation have led to a largely converging and increasingly detailed atomistic picture of the steps leading to the formation of the O-O bond, but the events at the active site after the release of the O₂ molecule have remained largely obscure, as the authors convincingly argue.

Due to the importance of the research topic and the targeted knowledge gaps, this computational study is potentially very well suited for publication in Nature Communications. However, there are unclear aspects of this work and critical issues as described below. These need to be addressed convincingly in the revised article and SI; this will likely require the submission of additional material. Ultimately, the article may represent a significant step forward worthy of publication in Nature Communications.

- (1) How were the coordinates for the Im0-O₂ intermediate obtained? What is the evidence that these coordinates are meaningful? The short lifetime in the ab-initio MD simulations imply that equilibration of the starting coordinates was not trivial. Could these coordinates correspond to an artificial high-energy state that is never populated in the real-world system?
- (2) Along the same line, the energy drop from the initial state to the Im3 state is about 30 kcal/mol. Such an enormous loss of free energy in this single step of photosynthetic oxygen evolution seems clearly incompatible with the energetics of the donor-side reactions of PSII. Could this be related to an Im0-O₂ state that can never be populated because it is too high in energy?
- (3) The mechanism of insertion of two water molecules followed by release of one water molecule into the bulk is surprisingly complex and somewhat counterintuitive in its complexity. Other starting structures might favor other water binding pathways. What is the argument that a reasonably unique water binding scenario has been discovered?
- (4) How do the computational results of the present study relate to Capone et al (2021, Biochemistry)? This work is cited. However, it should be clarified where the present study differs and where it represents clear progress when compared to Capone et al.
- (5) It could be said more clearly how the proposed events relate to the proton that generally is assumed to be released in the water binding and the S₀-state restoration process. Was this process addressed computationally? How (or why not)?
- (6) Details on the reaction path in the MEP calculations need to be provided in a clear way (in particular for the calculations leading to Figure 3a). How was the final state of the MEP selected? Intuitively or by a specific computational approach?
- (7) The initial steps (Fig. 2) appear to occur essentially spontaneously, without any significant energy barrier; the energy barriers for the steps in Fig. 3 also appear to be small. Is the overall process of S₀ restoration assumed to occur in very short times, or is the proton release associated with a large energy barrier that slows the overall process? A discussion of this point would be useful as it relates to the chances of seeing intermediates in future crystallographic snapshot data.
- (8) Suggestion: The ab-initio MD simulations would become more valuable for readers by additional analyses. During the ab-initio simulations of Figure 2 atoms move suddenly, some

of them by several Angstrom. Since nuclear tunneling effects are not considered in the used simulation approach, these are continuous (classical) movements. It would be good to show in form of graphical presentations how fast selected water molecules, in particular W3 and W7, move along a specific path. Do they accelerate and then decelerate? Are collisions among atoms safely avoided (no significant mutual penetration of the ionic or Van-der-Waals radii)? What is the maximal velocity and kinetic energy of the ballistic movement of W3? Such an analysis could provide an additional interesting twist to discussion of the simulation results.

Further points.

- It will be good to reconsider parts of the discussion of reaction kinetics on pg. 13. and 14. “Furthermore, these species evolving on a picosecond timescale are easily to escape snapshots using several hundreds of microseconds as the time interval.” (pg. 13) Intermediates that evolve on a picosecond time scale will be clearly impossible to detect in experiment with microsecond resolution. What is meant? “Nevertheless, it is worth mentioning that there is a critical inconsistency between Bhowmick et al. and Greife regarding the timing of onset of O-O bond formation, i.e. within 500~1200 μ s versus \sim 2.5 ms after 3rd flash” (pg. 14). The argument is not easily understandable. According Greife et al, O-O bond formation can be modelled as a Markov process with exponentially changing populations (which is standard and perfectly reasonable) with a time constant of 2.5 ms. This implies that the process of O-O bond formation is *completed* in 63.2% (= 1/e) of the photosystems at 2.5 ms. What is the critical inconsistency regarding the *onset* of O-O bond formation? Possible inconsistencies between these two exceptionally important studies published back-to-back in Nature should be very carefully evaluated, for example by means of kinetic modeling, in a small paper devoted entirely to this point. The cursory discussion in the present paper could too easily be the source of confusion.
- In Figure 2a, it is not well visible which water molecules are moving and what is the path of their movement. A separate figure with larger atom labels, showing also some residues is recommended.
- In Figure S1, it is not visible which of the residues are included in the quantum region. Please improve the presentation, including labels for all included residues. Moreover, please say explicitly whether and where atoms at the margin of the quantum regions were at fixed positions. The same for Figures S6, S7, S8, and S10.
- Clearly more details on the reaction path in the MEP calculations (and the calculations itself) need to be provided in the SI.
- The SI also requires an improved layout before it can be considered being final.

Reviewer #2 (Remarks to the Author):

The paper analyzes the process of water binding to the Oxygen-Evolving Complex (OEC), after the S3 to S0 transition of the Kok cycle, using Ab Initio Molecular Dynamics (AIMD) simulations. Their model of the OCE is assumed to be depleted from O(5) and a water ligand to Mn(1) after oxygen evolution, as suggested by the oxo-oxyl coupling mechanism for O-O bond formation, so they analyze how open coordination sites of Mn would be refilled by water (W) molecules if the cluster were to be left after O2 evolution with Mn(4) in oxidation state III and 5 ligands, including a hydroxyl (OH-) ligand that could serve as a base. Under those assumptions, the simulations show that an incoming W3 could bind to Mn(1) and deprotonate by transferring a proton to W2, leading to a close-cubane pre-S0 state, which could transition to an open-cubane structure of the OEC. In short, the proposal of utilizing a close-cubane structure to facilitate the proton transfer from W3 is astute and innovative, making use of the dangling Mn. Moreover, we appreciate that the author verifies the feasibility of unbinding W2 and transitioning the close-cubane S0 state to the open-cubane S0 state. According to Fig. 4, the authors also assume that W2 remains a OH ligand throughout the catalytic cycle. Clearly, the calculations rely heavily on certain assumptions such as oxo-oxyl in O-O bond formation, and a fully dissociative (stepwise rather than a concerted approach) process of water insertion, and the deprotonated state of W2 to enable the deprotonation of the incoming water. However, it is not sufficiently clear whether those assumptions are consistent with experimental data. For example, the protonation state of W2 as a hydroxyl ligand throughout the catalytic cycle should exhibit distinct hyperfine coupling constants that have been measured by ENDOR EPR measurements. I recommend the authors should run those calculations and compare to experiments to support, or rule out, their proposed protonation states.

Beyond the validity of the assumptions of the model, and therefore the validity of the conclusions, it is not clear whether the observation of water binding to a model of the OEC depleted of ligands is surprising, or a significant observation. So, I do not see how these calculations provide a significant contribution to the field. Additionally, I do not see any broader implications of this study for a general audience unfamiliar with the intricacies of the OEC.

Other aspects of concern, include:

- (1) The barriers for W2 unbinding in im2, and W1 displacement to become the new W2 while W7 becomes the new W1 need to be analyzed. In that case, changes in the hydrogen bonding network associated with binding W7 to Mn4 as the new W1 need to be analyzed since apparently the authors are suggesting breaking the interaction between D61 and W1 during this process.
- (2) The pathway for W2 unbinding is also unclear. How is the transit of W2 to the bulk? Is there any evidence of the exit of W2 in any S0 structure?

Minor Suggestions:

1. In Figure 2a, the label "Im2" should be "Im1", and "Im3" should be "Im2".

Reviewer #3 (Remarks to the Author):

The manuscript entitled "Closing Kok's Cycle of Nature's Water Oxidation Catalysis" by Yu Guo, Lanlan He, Yunxuan Ding, Lars Kloo, Dimitrios A. Pantazis, Johannes Messinger, and Licheng Sun presents a DFT-based theoretical study describing the last events occurring in the catalytic cycle of Photosystem II, subsequent to the molecular oxygen evolution, leading to the restoration of the initial state of the cycle.

The Mn4Ca cluster placed in the catalytic core of the Photosystem II is responsible for the water splitting reaction in the natural oxygenic photosynthesis, occurring along the five steps (S0-S4) of the Kok cycle. In the past years, the events occurring between the S0 and S3 states have been extensively characterized experimentally as well as theoretically. Recently, new experiments have

been reported (Ref.9 and 12 in the manuscript) shading light on the S3 → S4 → S0 transition, in which a peroxide bond between two oxygen atoms is formed, a O₂ molecule is released from the cluster and a water molecule is inserted in the Mn₄Ca cluster to restore the S0 state. Still, these experiments did not provide information on the mechanism of water insertion and the S0 reconstitution subsequent to the molecular oxygen evolution. These events have been previously investigated theoretically in different works, suggesting either a stepwise occurrence of oxygen release and water insertion (see e.g. Ref. 20 and 26) or a concerted mechanism (see e.g. Shoji et al. *J. Phys. Chem. B* (2018) 122:6491). In this work, combining ab-initio MD simulations, DFT-based energy calculations on snapshots extracted from the ab-initio MD simulations, and MEP calculations, the mechanism of the S0 restoration occurring after the oxygen release (assuming the scenario of a stepwise occurrence of oxygen release and water insertion) was explored.

Differently from previous published theoretical works, the results reported in the present manuscript indicate that a water molecule bound to the calcium ion of the cluster (namely W3), is inserted in the cluster, binding the Mn1 ion and therefore originating a closed-cubane structure. In this conformation an additional water molecule (namely W7) binds the Mn4 ion. These two events occur spontaneously within 30 ps of MD simulations, indicating a more or less absence of an energy barrier. Subsequently, a water molecule detaches from the Mn4 ion and the closed-cubane structure is converted into the open-cubane structure by breaking a bond between Mn1 and W3 and forming a bond between Mn4 and W3, thus restoring the (experimentally known) open-cubane structure of the S0 state. This new proposed mechanism for the S4 to S0 transition appears plausible based on the presented calculations and imply the presence of reversible isomerization of the S0 state, in line with that already suggested for other states of the Kok cycle.

The article appears well written, highly significant and of broad interest. The standard of the calculations carried out in the present work is of high level as can be expected from these authors and appropriate for well describing the investigated system. However, I have some concerns that should be addressed before recommending the publication of the manuscript on Nature Communications.

1. The spontaneous insertion in the cluster of the W3 water molecule binding the Mn1 ion as well as the spontaneous binding of the W7 water molecule to Mn4, occurring both in 30 ps of ab initio MD simulations, could be in my opinion artificially facilitated by the absence of the surrounding water molecules and protein residues tightly H-bonded with water molecules present in the simulated system. The system simulated by the authors is relatively large (369 atoms) and they claim that such a dimension is enough to capture "the dynamic phenomena involved in water insertion". I'm not so sure that this is necessary true. In this sense, an explicit treatment of the environment around the simulated region by means of e.g. QM/MM description, could lead to different results. It could prevent the spontaneous binding of the above mentioned water molecules associated to the relocation of other water molecules (see e.g. the binding of W7 to Mn4 with the relocation of W8 placed in a water channel truncated in the simulated system). Also, at pag. 10, "This is not a model artifact..." sounds a bit too strong to me. How can this possibility be totally excluded? Similar mechanisms known to occur in organic metallic chemistry can corroborate the findings of the authors, but not exclude the presence of artifacts. This point should be addressed by the authors, either explaining convincingly why their treatment/model should surely avoid artifacts or pointing out the limits of their calculations.

2. If I've understood correctly, the energies reported in Fig. 2c have been calculated on single snapshots of ab initio MD simulations, I mean single point calculations. If I'm right, I would suggest to repeat the calculations on different snapshots (e.g. for Im2 state not just the last snapshot but 10 snapshots taken from the last ps of simulation at interval of 0.1 ps, and similarly for Im0 and Im1) and provide an average value with a standard deviation, being the kind of energy calculation extremely sensitive to the particular geometry chosen for the calculation.

3. The authors used IRC analysis to calculate MEP. I think that, considering the broad audience of the journal, it would be useful to better explain (in the supporting information) the general scheme behind such a calculation. Advantage and disadvantage (if present) between potential energy surface walking methods and Chain-of-states methods (as NEB, already employed in the

investigation of the same mechanism studied here) could be also reported. I would also report in the method section in the main text where the starting geometries for the three reported MEP calculations (Asp61 protonation, water detachment, and closed-open cubane transition) come from.

Minor points:

1. The work of Shoji et al. J. Phys. Chem. B (2018) 122:6491, cited in the Supp. info, in my opinion should be cited in the main text at the end of pag. 5.
2. In Fig. 2a, Im2 should be Im1 and Im3 should be Im2.
3. In the caption of Fig.2, last sentence (Note that...): the reason because the three states are stationary points should be motivated, since from Fig.2 this is not clear being reported just the relative energies of the three states.
4. At Pag.10 "the W2 decoupling...endothermic by ca. 2-3 kcal/mol", being in the figure reported the Gibbs free energy should be "endergonic"

Overall, I found this work particularly significant and original for the field and I endorse the publication of the present manuscript after the above points will be addressed.

Reviewer #4 (Remarks to the Author):

The manuscript "Closing Kok's Cycle of Nature's Water Oxidation Catalysis" calculated the reset process of Mn₄CaO₅ (6) cluster during the final S₃→(S₄)→S₀ transition with molecular dynamics simulations combined with density functional calculations. The results suggest a likely missing link for closing the Kok's cycle. The structural isomerism in the S₀ state is proposed theoretically. The topics are crucial in understanding the nature's water oxidation catalysis, and the results are very interesting and inspiring for the design of artificial water oxidation catalysts. I recommend the publication of the present manuscript in Nature Communications with the following issues well addressed.

1. Although the structure of S₄ is claimed to be Mn(IV)-O● radical (Ref 19), the peroxide intermediate formed by oxo-oxyl coupling has not yet been confirmed experimentally. The different structure of the peroxide intermediate may form different structure after releasing O₂, which may influence the following reset process of Mn₄CaO₅ (6) cluster. This should be considered and added.
2. As calculated in Ref 19, the single-electron—multi-proton step facilitates the critical step of Mn(IV)-O● formation, which relies on a specific location of water molecules and hydrogen-bond interaction. In principle, the specific location of water molecules and hydrogen-bond interaction should play more crucial roles in the reset process of Mn₄CaO₅ cluster. This should be studied and discussed in detail.
3. As studied in Ref 9, the structural changes from 1,200 to 4,000 μs indicates that O₂ release and refilling of the cluster by bulk water and resetting of the catalytic center occur over an extended timescale. But this manuscript suggest that the resetting of the catalytic center is at picosecond timescale. Why does this huge difference exist? If the ultrafast resetting indeed completes at picosecond timescale, what happens from 1,200 to 4,000 μs?

Point-by-point responses to reviewers

Reviewer #1 (Remarks to the Author):

The process of light-driven water oxidation by photosynthetic organisms is crucial to life on Earth; its byproduct dioxygen has shaped the atmosphere and geosphere. Recent advances in experimental and theoretical work on photosynthetic water oxidation have led to a largely converging and increasingly detailed atomistic picture of the steps leading to the formation of the O-O bond, but the events at the active site after the release of the O₂ molecule have remained largely obscure, as the authors convincingly argue. Due to the importance of the research topic and the targeted knowledge gaps, this computational study is potentially very well suited for publication in Nature Communications. However, there are unclear aspects of this work and critical issues as described below. These need to be addressed convincingly in the revised article and SI; this will likely require the submission of additional material. Ultimately, the article may represent a significant step forward worthy of publication in Nature Communications.

Response: Thank you very much for your understanding and positive comments on our work, and for so many valuable comments and suggestions to help improve the quality of our manuscript!

(1) How were the coordinates for the Im0^{-O₂} intermediate obtained? What is the evidence that these coordinates are meaningful? The short lifetime in the ab initio MD simulations implies that equilibration of the starting coordinates was not trivial. Could these coordinates correspond to an artificial high-energy state that is never populated in the real-world system?

(2) Along the same line, the energy drop from the initial state to the Im2 state is about 30 kcal/mol. Such an enormous loss of free energy in this single step of photosynthetic oxygen evolution seems clearly incompatible with the energetics of the donor-side reactions of PSII. Could this be related to an Im0^{-O₂} state that can never be populated because it is too high in energy?

Response: We greatly appreciate for these very good questions, as they have inspired us with meaningful reflections for a more rigorous and scientific presentation of our work! Since the first two points you raised are closely related, please allow us to integrate them. The original structure is fundamentally based on the room-temperature serial femtosecond crystallography of the S₃ state, which was taken from the second flash (200 ms) data from Ibrahim et al.¹ (PDB ID: 6W1V, monomer A). After model construction for the S₃ state, the corresponding S₄ state was optimized after removing one proton and one electron from Ox (assumed as a hydroxide in S₃). Then the Im0^{-O₂}

structure was produced and optimized by removal of O5 and Ox. Thus, from S₃ to S₄ to Im0^{-O₂}, we have followed a normal and logical procedure for producing the coordinates of the Im0^{-O₂} intermediate. Note that direct experimental observation for the Im0^{-O₂} state (as well as the S₄ and S₄-peroxide states) has not been realized so far, so that for a computational study we have to construct the model based on comprehensive consideration on all available knowledge in this field. Since the present study is focused on how the cluster would evolve after O₂ release, the available ‘evidence’ or strong support for the existence of the Im0^{-O₂} state is mainly reflected in two aspects: 1) O5 and Ox as substrates for O-O bond formation, and 2) the complete removal of O₂ from the cluster for the subsequent entrance of a water molecule, which have been more specially demonstrated in Supplementary Text 1 and Text 4, respectively. In this sense, our Im0^{-O₂} model, in terms of a cavity inside the Mn₄CaO₄ cluster, makes sense for the structural evolution during the S₄-S₀ transition after O₂ release. It is the cavity, which is formed by three unsaturated 5-coordinated Mn(III) centers, that motivates water insertion to refill the empty coordination sphere. This explains the short lifetime (ca. 7.5 ps for the octet/ $\alpha\alpha\alpha\beta$ spin state and ca. 4.5 ps for the doublet/ $\alpha\beta\alpha\beta$ spin state) of Im0^{-O₂} in the MD simulations which rapidly converts to Im1 (also short, ca. 5 ps) and then to Im2. Since the separate geometry optimization (energy minimization) on Im0^{-O₂} can lead to a stable local minimum with the original structural cavity well maintained, also with overall support from the current literatures, we think the Im0^{-O₂} state should be transiently populated in the real-world system, despite with a short lifetime.

The comparison of the free energy loss from Im0^{-O₂} to Im2 with the energetics of the donor side reactions (from H₂O to P₆₈₀) of PSII has motivated us to contemplate on the capabilities and limitations of our theoretical simulations when quantitative estimation compared to experimental measurements are discussed. Dau and coworkers have made significant contribution on the energetics and kinetics of the S-state transitions^{2,3}, and Messinger and coworkers have estimated the driving force for S₄→S₀⁴, based on which the energetics (in ΔG) for the donor side is approximately in the range of 10~20 kcal/mol, thus our calculated free energy loss from Im0^{-O₂} to Im2 (ca. 30 kcal/mol) seems overestimated. However, based on the present model for which other parts and events beyond the local structural evolution surrounding the cluster are not covered, it may be reluctant to draw a direct corresponding relationship for the obtained free energy loss from Im0^{-O₂} to Im2 with the energetics of the donor side reactions of PSII. While the energy loss mainly accounts for the thermodynamic feasibility of the facile conversion, the causes of the difference should be valued including but possibly not limited to the following aspects. 1) The thermodynamic effect of O₂ release, especially for the translational entropy amounting to as much as -12 kcal/mol^{5,6}, is not included in

the Im0^{-O2} model. Since it is a process of entropy increase (decreasing the degree of randomness), the actual Im0^{-O2} state should be energetically lower. 2) The hydration entropy effects of Ca and Mn4 are not well expressed in the computed energetics. It is roughly estimated that the amounts are 2~3 kcal/mol and 3~4 kcal/mol, respectively, with respect to hydration by a single water molecule at room temperature, using the data for experimental values on hydration entropy taken from Marcus⁷. Since hydration causes entropy decrease (increasing the degree of randomness), the actual Im2 state should be energetically higher. 3) The extensive rearrangement of the hydrogen-bonding network (HBN) in the protein matrix related to the last proton release to the lumen was not taken into account. According to Bhowmick et al.⁸, the HBN rearrangement may be the source of the extended timescale of the S₀ restoration observed by femtosecond X-ray crystallography. Greife et al.⁹ suggest marked entropic slowdown for the S₃-S₄ transition associated with the HBN rearrangement caused by proton release during the S₃-S₄ transition. While it is clear that the HBN rearrangement can slow down the overall kinetics by destabilizing the rate-limiting TS (by 6.5 kcal/mol entropic contribution), it may affect the thermodynamics of an intermediate or final state in a similar way. Since proton release is not completed in Im2, the above effect should be deducted when compared with the energetics of donor-side reactions. 4) The computed energetics do not fully represent the complete donor-side reactions, but only the intrinsic/local energetics of the conformational evolution occurring at/around the inorganic cluster. Limited by the model size and current computation capability, it is difficult to make a computational prediction on numerically how large an effect from other donor-side reactions would exert on the energetic difference between Im0^{-O2} and Im2; however, the above analyses show plausible reasons for the sources of the discrepancy for quantitative comparison with the energetics of the donor-side reactions, which, if possible to take into account somehow, would significantly narrow the gap. Consequently, while the above factors leading to the computational overestimate within the model imitations are beyond the scope of our simulations, caution should be given when comparing with experimental data. Finally, we emphasize the validity of the structural evolution of the Mn cluster from Im0^{-O2} to Im2 and that the resulting central conclusion remains unaffected. However, in order to avoid unnecessary misunderstandings, we decide to remove the energy profile from the main text but still keep the data in Supplementary Table S1.

(3) The mechanism of insertion of two water molecules followed by release of one water molecule into the bulk is surprisingly complex and somewhat counterintuitive in its complexity. Other starting structures might favor other water binding pathways. What is the argument that a reasonably unique water binding scenario has been discovered?

Response: Thank you for giving us a chance to explain. We cannot comment on whether the process is “unrealistically complex” or not, because the process reported here is the outcome of our simulations, and we are confident it represents a reasonable and feasible scenario and can be understandable seen from the following aspects. Although various starting structures might favor other water binding pathways, we assume there are two points for our starting structure whose rationality need to be stressed. The first one is the mechanism of O-O bond formation chosen, and the second one is the foundation of the Mn_4CaO_4 cluster with a structural cavity as the starting point. These have been demonstrated in Supplementary Text S1 and Text S4, respectively, and we believe the two bases for the starting point are solid for the following study.

Insertion of two water molecules (W5 to Ca and W7 to Mn4) is the special outcome of W3 binding to Mn1 instead of Mn4. First of all, the propensity of W3 entering the cavity of the cluster is thermodynamically driven by the more stabilized structure of the cluster from three unsaturated 5-coordinated Mn(III) (Mn1, Mn3 and Mn4) to saturated 6-coordination. Secondly, the reason why W3 binding to Mn1 first instead of Mn4 is attributed to their discrepant charge distributions (Mn1 more positive than Mn4), the steric effect of the slightly rotated W2 on Mn4 and the hydrogen bonding (HB) interaction of W2-W3 which hinders W3 directly binding to Mn4. W5, which was H-bonded to W3, is pulled by the W3 motion and then occupies the original position of W3 on Ca. W5 is influenced in such a way and is also reflected in literatures on the $\text{S}_2\text{-S}_3$ transition¹⁰. Thirdly, the subsequent W7 binding to Mn4 is a straightforward consequence of the ‘pivot/carousel’-like ligand reorganization around the dangler Mn4(III). During the MD simulation, it is found that the spontaneous W2 protonation from W3 causes further rotations of W2 and W1 and expansion of the $\angle\text{Mn3-O4-Mn4}$ angle, and creates a vacant coordination sphere on Mn4 toward the W7, which soon binds to Mn4 as its sixth ligand.

After formation of the pre- S_0 (W2-bound) state upon H^+ release in Im2, W2 dissociation is necessary for $\text{W3}(\text{OH}^-)$ transfer to Mn4 to form the open-cubane S_0 state (that has been experimentally identified). Herein, the dissociation of $\text{W2}(\text{H}_2\text{O})$ at this state is not groundless but being a chemically sensible process. The transfer of W2 to the bulk is, in terms of behavior, very similar to the last step of the proposed mechanism of water exchange for the OEC. According to Fig. 4 in the revised manuscript, a low-barrier route for $\text{W2}(\text{H}_2\text{O})$ de-coordination and release from Mn4(III) has been located. The obtained barrier height for W2 detaching from Mn4(III) is quite similar to a recent report on a biomimetic polyoxometalate water oxidation catalyst¹¹. In the model calculation, the Mn4-W2 distance is changed from 2.18 Å (W2 bound) to 3.99 Å (W2 unbound). A distance of ~ 4 Å is generally considered in the water exchange region, and in any case, to move the water even further out costs very little energy. Due to the

size limitation of the model, only a finite value $\sim 4 \text{ \AA}$ of the Mn4-W2 distance can be obtained but this distance adequately illustrates its tendency to transit to the bulk. The situation and explanation are in analogy to substrate water exchange in the S_1 , S_2 , and S_3 states proposed by Siegbahn¹². For the S_0 state, there is still no computational studies on the water exchange mechanism, as well as the exit of W2 in any S_0 structure in literatures, but essentially, they all share the common point that Mn4(III) in a closed-cubane structure serves as the station for water dissociation. The reactivity is enabled by the presence of Jahn-Teller (J-T) effect at Mn(III) which extends the ligand bonding distance and thereby facilitates its de-coordination. Furthermore, in a scheme from the Messinger group based on the experimental data (in their Scheme 3), W2 unbinding from Mn4 is involved in a closed-cubane structure in their hypothetical water exchange mechanism for the S_0 state¹³. That means W2 release from Mn4(III) from the cluster makes sense in such circumstance. These have been reflected in the main text and Supplementary Text S6 and Text S7. Anyway, the topic and the details are still contentious, and the present work serves to “flesh out” currently the most plausible scenario that remains under discussion.

(4) How do the computational results of the present study relate to Capone et al. (2021, Biochemistry)? This work is cited. However, it should be clarified where the present study differs and where it represents clear progress when compared to Capone et al.

Response: Thank you for your good question related to this important publication. For the past years, Guidoni and coworkers have made very influential work to this filed by means of theoretical calculations mainly using QM/MM MD, which has inspired us a lot. Previous to our present study, Capone et al. also looked into the mechanism of oxygen evolution and Mn_4CaO_5 cluster restoration¹⁴, in which the major contributions mainly include two points: 1) confirmation of the Ca-bound W3, instead of Mn4-bound W2, as the water inserting into the cavity formed by O_2 release; 2) testification for the validity of the two-step (stepwise) mechanism for O_2 release and water (W3) insertion against the concerted mechanism. For our present study, the observation that W3 as the inserted water clearly agrees well with them, and our starting structure with the complete removal of the O_2 molecule is also consistent with their finding which is also referred as the foundation, also in line with Siegbahn’s previous results¹⁵. For reconstitution of the Mn_4CaO_5 cluster, while Capone et al. assume W3 (along with deprotonation to W2) would directly move to the bridge position between Mn3 and Mn4 and form the open-cubane structure of the S_0 state, we propose a different pathway for how W3 should enter the cavity, i.e. W3 binding to Mn1 first for formation of the closed-cubane intermediates along with W3 deprotonation to W2, which triggers the ‘pivot/carousel’-like ligand reorganization around Mn4 and W7 binding to Mn4, and then the open-cubane structure of the S_0 state would form after W2(H_2O) dissociation

and W3(OH⁻) transfer to Mn4. Regarding the methodology, Capone et al. used QM(DFT)/MM MD(Car-Parrinello) for 10 ps while we used *ab initio* (DFT) MD(Born-Oppenheimer) for 30 ps which observes the spontaneous W3 binding. We are not so sure if the methodological differences have large effects (without recommending or criticizing one over the other), but Capone et al. did not test the possibility proposed here and just conducted W3 (also for W2) move straightforward to bind to Mn3/Mn4 in their directed MEP calculations. In their MD simulation within 10 ps, they indeed found the predisposition of W3 moving to the vicinity of the cavity, but it is uncertain whether the further movement of W3 onto a specific Mn (toward Mn1 or Mn4) could be observed if the simulation time were extended. Finally, we emphasize equal importance for the two plausible routes, without excluding the possibility of the old scenario, and further comparative studies may be needed. However, while the target structure is the same, we present possible intermediates in closed-cubane structures *en route* to reach the S₀ state, which are important for understanding the structural flexibility prevailing in the cycle. This has been concisely added in the Discussion part of the main text and in detail in Supplementary Text S18.

(5) It could be said more clearly how the proposed events relate to the proton that generally is assumed to be released in the water binding and the S₀-state restoration process. Was this process addressed computationally? How (or why not)?

Response: Thank you for giving us a chance to clarify. The process was addressed computationally by MEP calculations (Supplementary Fig. S6 and Table S2) only for W1 deprotonation to Asp61 in the Im2 state (the green arrow in Im2 in Fig. 2), and the results indicate the deprotonation process is quite facile. For the following reasons, we consider W1(H₂O) should be the deprotonation site which releases H⁺ to lumen *via* Asp61: 1) W1 becomes the new W2 in Im2 since it is translocated to the *trans* position of O4 by the ‘pivot/carousel’-like ligand reorganization; 2) W2 is generally considered as a hydroxide for the metastable forms of the S states¹⁴⁻²⁰; 3) W1 is in strong hydrogen-bonding interaction with Asp61 which is generally recognized as the gate for releasing proton^{8,9,15,19,21-25}. Further proton release from Asp61 to other groups and then to bulk along the proton channel is assumed feasible and was not investigated because 1) the function for Asp61 releasing H⁺ has been widely acknowledged (see ref above); 2) the rearrangement of HB network in the proton channel should cause little effect on the structural evolution of the Mn cluster; 3) the residues and waters in the proton channel are not sufficiently included in the model setup since it is not the focus of this study. This has been added as Supplementary Text S9. By the way, as shown above, this can be indeed one reason why the computed energetics from Im0^{-O₂} to Im2 do not represent the full donor-side reactions in PSII.

(6) Details on the reaction path in the MEP calculations need to be provided in a clear

way (in particular for the calculations leading to Figure 3a). How was the final state of the MEP selected? Intuitively or by a specific computational approach?

Response: Thank you for giving us a chance to clarify. Regarding how was the final state of the MEP selected, of course not intuitively. Take Figure 3a (now Figure 4) as an example, the general scheme is 1) locate the TS for W2 dissociation from Mn4; 2) run the frequency analysis and check the imaginary vibration mode; 3) run IRC analysis based on the Hessian of the TS which trace the reaction path to pre-S₀(W2-bound) and pre-S₀(W2-unbound); 4) optimize pre-S₀(W2-bound) and pre-S₀(W2-unbound); 5) make ZPE and thermal corrections on them on the MEP. Locating the TS in the first step is the most critical and challenging. As we do not know the Mn4-W2(O) distance in the TS, we performed a relaxed PES scan by smoothly increasing the Mn4-W2(O) bond length with a step size of 0.05 Å, and picked the point with highest energy and lowest gradient as an approximation (initial guess structure) for the subsequent TS optimization (by Berny algorithm). Then following the above procedure, the final state of the MEP can be located whose Mn4-W2(O) distance is about 4 Å. Note that ideally in the real system, the distance should be much longer and uncertain because it would be highly dissociative in the bulk; however, practically in a cluster model calculation, such a distance can be considered adequate because to move the water even further out costs very little energy. This is also in good agreement with the value of the dissociated water released into the bulk shown in the mechanism of substrate water exchange¹². Regarding the details on the reaction path in the MEP calculations, we assume you refer to the ‘intermediate images’ (along the reaction coordinate with changes of their energies, key distances and spin populations, etc.) positioned between the initial and final states, like what are usually shown by the NEB method. NEB optimizes a series of intermediate images positioned between the initial and final states of the reaction. These images are evenly spaced along the reaction path and are optimized to find the lowest energy configurations while maintaining a fixed separation from their neighboring images. However, differently, we use the IRC method for calculating the MEP which is defined as the steepest-descent path on the PES from the TS down towards a local minimum, i.e. reactant (Rea) and/or product (Pro). So, there are no intermediate images needed for locating TS. Instead, after TS has been correctly located, the IRC analysis will trace the reaction path to the corresponding reactant and product (which should be optimized again). As the points produced between TS and Rea/Pro on the IRC curve are not stationary points and do not undergo optimization, it seems unnecessary to show them on the Gibbs free energy profiles. However, Supplementary Figs. S12-S17 have been provided for the IRC curves with details of the changes regarding the electronic energies (without ZPE and thermal corrections), RMS gradient norm and key bond lengths along the IRC. Spin populations are not shown because

there are no obvious variations on Mn or the ligands, and Mn1(III)Mn2(IV)Mn3(III)Mn4(III) remains throughout. The calculation methodology and general scheme specially compared to NEB have been shown in Supplementary Text 15.

(7) The initial steps (Fig. 2) appear to occur essentially spontaneously, without any significant energy barrier; the energy barriers for the steps in Fig. 3 also appear to be small. Is the overall process of S₀ restoration assumed to occur in very short time, or is the proton release associated with a large energy barrier that slows the overall process? A discussion of this point would be useful as it relates to the chances of seeing intermediates in future crystallographic snapshot data.

Response: Thank you for your good question, and you have raised a very important point. For a quick answer to your question, we speculate the latter possibility, i.e. proton release associated with a relatively larger energy barrier could be in part responsible for slowing the overall process. Here we say, ‘relatively larger’ (not large), because the overall process is still in millisecond timescale. We also emphasize that the events we have simulated do not cover the whole process of the S₀ restoration, which, apart from the O₅ recovery, may include kinds of structural rearrangements (possibly and partially caused by proton release, as you have indicated) in the protein environment far away from the Mn₄CaO₅ cluster which cannot be considered in our model and simulations. Bhowmick et al.⁸ observed that from 1,200 to 4,000 μs several structural changes occur, and most of these changes are indicative of O₂ release and/or water insertion. This indicates that O₂ release and refilling of the cluster by bulk water and resetting of the catalytic center occur over an extended timescale. Within this period, 1,200 and 2,000 μs are the two essential timepoints that are closely related to our work. According to their identification, the 1,200 μs snapshot signifies the onset of O₂ evolution; the 2,000 μs snapshot, without Ox on the electron density omit map, indicates completion of binding of a water that refills the vacant site formed by O₂ release. On this basis, the process we have suggested in the present study should in principle transiently reside between these two timepoints, but being a very short period because of the picosecond timescale for water insertion and nanosecond timescale for the subsequent closed-to-open- cubane transformation of the cluster. It is understandable that the ultrafast conversion cannot be captured by the XFEL crystallography with hundreds of microseconds as the interval. According to Bhowmick et al.⁸, additional distances changes between Y_Z and His190 are observed between 2,000 and 4,000 μs, which may be due to rearrangement of the HB network related to the last proton release but are not well understood currently. In our proposal, H⁺ is released from W1 (as the new W2) to the lumen *via* Asp61 in Im2, and this is assumed to be the second proton release during the S₃-S₀ transition. After gated by Asp61, the released H⁺ would pass by substantial

water and protein residue groups located in the proton channel of PSII which may cause pronounced variations in rearrangement of the HB network (including water-water, protein-protein, and water-protein interactions) coupled to protein dynamics²⁶ and possibly other unknown structural changes, together responsible for the extended timescale between 2,000 and 4,000 μ s. For example, the increase of the Y_Z-D1-H190, Ca-D1-E189 and Mn4-O5 distances and decrease of the Mn1-Mn4 distance were observed. As you have indicated, there could be chances of seeing intermediates during the extended timescale for future crystallographic snapshot data. Thus, we clarify that neither we suggest the resetting of the catalytic center is at picosecond timescale nor our suggested process covers the whole period from 1,200 to 4,000 μ s; but rather the mechanism/progression suggested by our MD simulations and MEP calculations represents only an ultrashort phase (from picosecond to nanosecond timescale) embedded in the timepoints between 1,200 (precisely some timepoint after this when O₂ has been released) and 2,000 μ s. Besides, it is noted that in Bhowmick et al.'s definition, the final S₀ state is assigned to 3F(200 ms); however, by the 2,000 μ s timepoint after 3F, the O5 omit map density is restored considerably compared with the S₃ and S₀ states, which indicates water insertion after O₂ release has occurred, albeit not yet the final S₀ state with fully restored electron densities and other qualified indicators. In our modelling and definition, the final S₀ state (i.e. the S₀^A state) is reached as long as the open-cubane structure is formed within the Mn₄CaO₅ cluster, which should correspond to a timepoint close to 2,000 μ s. So, there is a semantic discrepancy on the definition of the S₀ state between Bhowmick et al.'s and ours, but both share the most important point that O5 has been restored between Mn3 and Mn4 as the basic feature of the cluster in the S₀ state. Since our model and simulations cannot and do not aim to represent other structural changes after 2,000 μ s toward the S₀(3F(200 ms)) state, the above-mentioned cautions should be taken when compared to the experimental findings regarding the extended timescale. These have been clarified concisely in the Discussion section in the main text and in detail in Supplementary Text S16.

(8) Suggestion: The ab initio MD simulations would become more valuable for readers by additional analyses. During the ab initio simulations of Figure 2 atoms move suddenly, some of them by several Angstroms. Since nuclear tunneling effects are not considered in the used simulation approach, these are continuous (classical) movements. It would be good to show in form of graphical presentations how fast selected water molecules, in particular W3 and W7, move along a specific path. Do they accelerate and then decelerate? Are collisions among atoms safely avoided (no significant mutual penetration of the ionic or Van-der-Waals radii)? What is the maximal velocity and kinetic energy of the ballistic movement of W3? Such an analysis could provide an additional interesting twist to discussion of the simulation results.

Response: Thank you for your interesting and instructive idea. Accordingly, we have collected all velocities and kinetic energies of W1, W2, W3, W5, W6, W7, W8 (represented by the oxygen atoms) and the interatomic distances of W3-W5, W5-W6, and W7-W8 (also represented by the oxygen atoms) for every step during the simulations for both the octet/ $\alpha\alpha\alpha\beta$ and doublet/ $\alpha\beta\alpha\beta$ spin states, as shown in the figures below. Collisions among the inspected atoms are safely avoided because all the interatomic distances remain above 2.5 Å. The velocities basically fluctuate between 1~15 Å/ps (mainly 3~10 Å/ps), and the kinetic energies between 0.1~4 kcal/mol (mainly 0.1~2 kcal/mol). Unfortunately, we cannot extract information on when W3 and W7 accelerate/decelerate, or pinpoint any special timepoint when the ballistic movement of W3 has maximal velocity and kinetic energy, because the fluctuations are continuous and almost broadly uniform. In other words, the expected acceleration and deceleration are drowned in the statistical velocity ‘noise’, so that the proposed properties cannot be extracted from the simulations. However, the motion states of particles can be qualitatively reflected from the interatomic distances already shown in Fig. 3, i.e. distance variation per unit time. For example, W3 (the red line) should have relatively larger velocities or kinetic energies in the yellow phase, i.e. ca. 8~14 ps for the octet/ $\alpha\alpha\alpha\beta$ spin state and ca. 4~12 ps for the doublet/ $\alpha\beta\alpha\beta$ spin state; W7 (the blue line) should have the largest average velocities or kinetic energies during the transition period from the yellow phase to blue phase, i.e. ca. 13~16 ps for the octet/ $\alpha\alpha\alpha\beta$ spin state and ca. 10~13 ps for the doublet/ $\alpha\beta\alpha\beta$ spin state.

Further points:

It will be good to reconsider parts of the discussion of reaction kinetics on pg. 13. and 14. “Furthermore, these species evolving on a picosecond timescale are easily to escape the snapshots using several hundreds of microseconds as the time interval.” (pg. 13) Intermediates that evolve on a picosecond time scale will be clearly impossible to detect in experiment with microsecond resolution. What is meant?

Response: Thank you for your question. This is precisely what is meant as you have correctly understood that the intermediates that evolve on a picosecond time scale will be clearly impossible to detect in experiment with microsecond resolution. We just emphasize that this is a possible reason why the femtosecond X-ray crystallography is not able to capture our proposed intermediates, so the non-observation in experiment does not necessarily exclude the validity of the mechanism. Anyway, as the previous sentence has described the situation well, we have deleted the sentence to avoid overemphasizing the meaning.

“Nevertheless, it is worth mentioning that there is a critical inconsistency between Bhowmick et al. and Greife regarding the timing of onset of O-O bond formation, i.e. within 500~1200 μ s versus ~2.5 ms after 3rd flash” (pg. 14). The argument is not easily understandable. According Greife et al, O-O bond formation can be modelled as a Markov process with exponentially changing populations (which is standard and perfectly reasonable) with a time constant of 2.5 ms. This implies that the process of O-O bond formation is completed in 63.2% ($=1-1/e$) of the photosystems at 2.5 ms. What is the critical inconsistency regarding the onset of O-O bond formation? Possible inconsistencies between these two exceptionally important studies published back- to-back in Nature should be very carefully evaluated, for example by means of kinetic modeling, in a small paper devoted entirely to this point. The cursory discussion in the present paper could too easily be the source of confusion.

Response: Thank you for your explanation and correction. We just simply pointed out the discrepancy between ‘500~1200 μ s’ and ‘2.5’ ms after 3F for the timing of onset of O-O bond formation. We now realize such a statement here is not appropriate in this manuscript according to your explanations. It is absolutely beyond our scope and intentions to analyze independently the data in these two exceptionally important studies. We agree with you and have deleted this sentence.

In Figure 2a, it is not well visible which water molecules are moving and what is the path of their movement. A separate figure with larger atom labels, showing also some residues is recommended.

Response: Thank you for your good suggestion. These have been revised in Fig. 2.

In Figure S1, it is not visible which of the residues are included in the quantum region. Please improve the presentation, including labels for all included residues. Moreover, please say explicitly whether and where atoms at the margin of the quantum regions were at fixed positions. The same for Figures S6, S7, S8, and S10.

Response: Thank you for your good suggestion. Actually, all the residues are included in the quantum region described by DFT (in full quantum mechanical treatment), as shown in both Introduction and Methods sections. Backbone constraints by fixing α -carbons of peptide bonds were applied throughout, since they are the most accurate positions determined by the X-ray analysis²⁷. The fixed atoms have been explicitly shown in the revised Fig. S1 and the Figs. S6, S7, S8, and S10 have been also adjusted accordingly.

Clearly more details on the reaction path in the MEP calculations (and the calculations itself) need to be provided in the SI.

Response: Thank you for your suggestion. Supplementary Figs. S12-S17 have been provided for the IRC curves with details of the changes regarding the electronic energies (without ZPE and thermal corrections), RMS gradient norm and key bond lengths along IRC. For the calculation itself (the general scheme), Supplementary Text 15 has been added.

The SI also requires an improved layout before it can be considered being final.

Response: Thank you for your care. We have checked the format and improved the layout with a detailed ‘Contents’ in the front page for supplementary texts, tables, figures, references. Other things like tables that go over a page have been fixed.

Reviewer #2 (Remarks to the Author):

The paper analyzes the process of water binding to the Oxygen-Evolving Complex (OEC), after the S_3 to S_0 transition of the Kok cycle, using Ab Initio Molecular Dynamics (AIMD) simulations. Their model of the OCE is assumed to be depleted from O(5) and a water ligand to Mn(1) after oxygen evolution, as suggested by the oxo-oxyl coupling mechanism for O-O bond formation, so they analyze how open coordination sites of Mn would be refilled by water (W) molecules if the cluster were to be left after O_2 evolution with Mn(4) in oxidation state III and 5 ligands, including a hydroxyl (OH-) ligand that could serve as a base. Under those assumptions, the simulations show that an incoming W3 could bind to Mn(1) and deprotonate by transferring a proton to W2, leading to a close-cubane pre- S_0 state, which could transition to an open-cubane structure of the OEC. In short, the proposal of utilizing a close-cubane structure to facilitate the proton transfer from W3 is astute and innovative, making use of the dangling Mn. Moreover, we appreciate that the author verifies the feasibility of unbinding W2 and transitioning the close-cubane S_0 state to the open-cubane S_0 state. According to Fig. 4, the authors also assume that W2 remains a OH ligand throughout the catalytic cycle. Clearly, the calculations rely heavily on certain assumptions such as oxo-oxyl in O-O bond formation, and a fully dissociative (stepwise rather than a concerted approach) process of water insertion, and the deprotonated state of W2 to enable the deprotonation of the incoming water. However, it is not sufficiently clear whether those assumptions are consistent with experimental data. For example, the protonation state of W2 as a hydroxyl ligand throughout the catalytic cycle should exhibit distinct hyperfine coupling constants that have been measured by ENDOR EPR measurements. I recommend the authors should run those calculations and compare to experiments to support, or rule out, their proposed protonation states.

Response: Thank you very much for your careful review and raising important points regarding the assumptions made in our study. It is indeed crucial to assess the consistency of these assumptions with experimental data to validate the proposed protonation states and the overall model. Please allow us to note that for such a highly complex system with many unsolved/debatable scientific issues, a computational modelling work for a specific topic has to be constructed based on certain assumptions, as long as they are reasonable and well founded. Firstly, for the mechanism of O-O bond formation, although the bonded O-O moiety (peroxide) has not been directly observed experimentally yet (mainly referring to X-ray crystallography), the preponderance of available evidence from most experimental and theoretical studies, including XFEL²⁸, time-resolved spectroscopies⁹, isotope labeling water exchange experiments²⁹⁻³¹, DFT modelling^{15,32-35} and large-scale QM/MM simulations³⁶⁻³⁹, etc., strongly supports the oxo-oxyl coupling mechanism (at least for

O5 and Ox as substrates). Also, both of the two recent breakthroughs by Bhowmick et al.⁸ and Greife et al.⁹ (as the background of our present study) have recommended it as the most viable or even compelling pathway, respectively. Secondly, for the fully dissociative process of water insertion, Capone et al.¹⁴ have specially compared the stepwise and concerted pathways of O₂ release and water insertion by QM/MM MD simulations, and clearly favor the stepwise over the concerted mode⁴⁰, the latter of which is energetically prohibited by the high barrier. The stepwise mode is also consistent with Siegbahn's calculations on the S₄-S₀ transition¹⁵, for which the obtained energetics agrees well with experimental observations. Thirdly, it is true that different protonation patterns lead to clearly discernible spin projections and ⁵⁵Mn hyperfine coupling parameters. For the protonation state of W2, although the W2=H₂O scheme cannot be ruled out^{24,34,41-43}, there have been a majority of research work and review papers using the W2=OH⁻ scheme^{12,13,15,19,20,29,32,33,36,44-54}. Even for Greife et al.'s theoretical section⁹, the core proposal of single-electron—multi-proton transfer event relies heavily on the hydroxide nature (OH⁻) of W2 for accepting the proton abstracted from O6. We appreciate your valuable suggestion about calculations on hyperfine coupling constants for W2=H₂O or OH⁻, however, such calculations are already covered in literatures with respect to the lower S-states where ENDOR EPR experiments have been conducted.^{16-18,45,46} No experiments are available for the higher S-states and certainly not for intermediates that cannot be observed spectroscopically past the S₃ state. Here we list two specific examples by partial authors of our present manuscript supporting W2=OH⁻. Ames et al.'s BS-DFT calculations¹⁸ suggest that one of the two water molecules (assigned to W2) that are proposed to coordinate to the outer Mn4 of the cluster is deprotonated in the S₂ state, as this leads to optimal experimental agreement, reproducing the correct ground state spin multiplicity (S=1/2), spin expectation values, and EXAFS-derived metal-metal distances; Krewald et al.¹⁷ have made comprehensive evaluations on considerable models with different oxidation and protonation states with respect to their geometric, energetic, electronic, and spectroscopic properties compared to available experimental EXAFS, XFEL-XRD, EPR, ENDOR and Mn K pre-edge XANES data. While the high-valent scheme is conclusively favored, W2=OH⁻ (called 'three-proton model' therein) remains the preferred description whereas W2=H₂O ('four-proton model') cannot fit both EPR signals of the S₂ state. Consequently, we are confident that we have made reasonable and valid assumptions of the model for the further modelling on the water insertion process. Without these assumptions, it would be very difficult to initialize the model system and carry out the following simulations towards the S₀ state after O₂ release. On the other hand, reassessment on these assumptions themselves would also be of great significance, but we think these special calculations on those aspects should belong to

a separate work, and we will look for a such a chance later on. Relevant contents have been reflected in Supplementary Text S1, Text S3 and Text S4.

Beyond the validity of the assumptions of the model, and therefore the validity of the conclusions, it is not clear whether the observation of water binding to a model of the OEC depleted of ligands is surprising, or a significant observation. So, I do not see how these calculations provide a significant contribution to the field. Additionally, I do not see any broader implications of this study for a general audience unfamiliar with the intricacies of the OEC.

Response: Thank you for giving us a chance to clarify. Based on the aforementioned rationality of O5 and Ox as substrates and the fully dissociative process of water insertion, it is natural and logical to expect the observation of water binding as meaningful occurrence. The point of the work is not only that water will eventually bind to the unsaturated coordination of Mn in the structural cavity, but also the overall sequence and coordination of events after O₂ release based on a starting point that is rarely specially studied computationally before the present work. As such, our study reports a new mechanistic scenario that embodies a previously unexplored important link to recovery of the catalyst in natural photosynthetic O₂ evolution, which we believe is worth serious consideration in the future. It is important to note that scientific research often focuses on specific aspects of a large field, and the significance of a study to readers can vary depending on the context and target audiences. In this case, our study is more relevant and valuable to researchers and experts in the field of the OEC and its role in photosynthesis. The observation of water binding to a model of the OEC depleted of ligands could be significant within the specific context of understanding the behavior and dynamics of the OEC. By studying how water interacts with the OEC, researchers can gain insights into the mechanism of cluster evolution after O₂ release and potentially uncover new information about the functioning of this crucial biological system. These calculations might contribute to the ongoing efforts to develop more efficient artificial photosynthetic systems or improve our understanding of natural photosynthesis. While the study may not have immediate broader implications for a general audience unfamiliar with the intricacies of the OEC, it is important to note that scientific discoveries often build upon each other, and seemingly small findings can eventually contribute to broader advancements. Additionally, studies like this help in expanding our fundamental understanding of biological systems, which can have long-term implications for various applications and technologies. It is important for researchers to communicate their findings effectively to both specialized and general audiences. By providing clear explanations and highlighting the potential impact of their work, researchers can bridge the gap between technical details and broader significance, making their research accessible to a wider audience. Below we copied

from <https://www.nature.com/ncomms/submit/guide-to-authors> to show that *Nature communication* welcomes publications of high relevance for a specific field (here: water oxidation in natural and artificial photosynthesis): “*Nature Communications is an open access, multidisciplinary journal dedicated to publishing high-quality research in all areas of the biological, physical, chemical and Earth sciences. Papers published by the journal represent important advances of significance to specialists within each field.*”

Other aspects of concern, include:

(1) The barriers for W2 unbinding in Im2, and W1 displacement to become the new W2 while W7 becomes the new W1 need to be analyzed. In that case, changes in the hydrogen bonding network associated with binding W7 to Mn4 as the new W1 need to be analyzed since apparently the authors are suggesting breaking the interaction between D61 and W1 during this process.

Response: Thank you for your good suggestions. The barrier for W2 unbinding in Im2 (precisely in the pre-S₀ state with deprotonated W1) is calculated to be 5.0 and 4.0 kcal/mol for the octet/ $\alpha\alpha\alpha\beta$ and doublet/ $\alpha\beta\alpha\beta$ spin states, respectively, as shown in Fig. 4 of the revised manuscript. W2 unbinding is enabled by the presence of Jahn-Teller effect at Mn4(III), which can extend Mn-ligand bonds and thereby lower reaction barriers for bond dissociation. The obtained barrier height is similar to the case in Schwiedrzik et al.'s study on a biomimetic polyoxometalate water oxidation catalyst¹¹, in which 5.8 kcal/mol was obtained for ligand dissociation. We have added relevant words therein.

W1 displacement to become the new W2 while W7 becomes the new W1 are considered as barrierless (or almost) processes since they were observed in our MD simulations to take place spontaneously and complete within only 1~2 picoseconds in the transition region from Im1 to Im2; furthermore, they are actually synchronously coupled and show the ‘pivot/carousel’-like ligand reorganization around the dangler Mn4, together with W2 rotation after its protonation. The ‘pivot/carousel’ mechanism was proposed by Retegan et al.⁴⁷, Askerka et al.⁵⁵⁻⁵⁷, and Capone et al.⁵⁸ on the S₂-S₃ transition, and the barrier for water binding to Mn4(IV) (from the O4 channel) was reported to be 4.5 (Retegan et al.) and 8 kcal/mol (Capone et al.) (no barrier data for Askerka et al.), which are sufficiently low to occur but still kinetically slower than the process focused on here. We surmise the more facile ligand reorganization suggested here could be attributed to two factors that are absent in the ‘pivot/carousel’ mechanism for S₂-S₃: valence (III) of Mn4 and the W3-W2 HB interaction. The bond strength of ligands (H₂O) on Mn4(III) should be less than that of OH⁻ on Mn4(IV), so that ligand displacement around Mn4(III) can be easier. Besides, W2 is connected by the strong H-bond interaction to W3 which is bonded to Mn1, making W2 rotate some angle around

Mn4 toward the cavity in Im1. W1 is also affected and moves closer to the para-position of O4 for a more stabilized geometry and thus create enough room to accommodate W7. W2 protonation from W3 weakens the ‘structural *trans* effect’⁵⁹⁻⁶² of Mn4 and significantly facilitates W7 binding. These are supposed to account for the smooth and barrierless translocations of water ligands on Mn4(III) observed in the MD simulations. Note that because Mn4 moves outwards during this process, the positions of the new W2 and W1 in Im2 somewhat deviate from those of the original ones in Im0^{-O2} (but will return to normal in S₀^A). In this process, the H-bond interaction between W1 and D61 is actually not broken because the cluster is expanded by the elongated Mn1-Mn4 distance (or by enlarged \angle Mn3-O4-Mn4) and the W1-D61 interaction is maintained (although W1 rotates some angle around Mn4) and this is another clear difference from the ‘pivot/carousel’ mechanism for S₂-S₃. W1, to be the new W2 (as it is *trans* to O4 in Im2), is H-bonded to carboxyl oxygen of Asp61 and amide oxygen of Asp170 throughout. In our proposal, Asp61 will abstract one proton from the new W2 and release to lumen to become the pre-S₀ state. It is expected that after proton donation from the new W2, the location of Asp61 would be quite flexible and the side chain would largely rotate, as seen in the classical MD simulation by Rivalta et al.⁶³ and *ab initio* QM/MM MD simulation by Narzi et al.⁵³, which would then break the H-bond between them. W7 is H-bonded to O4, W8, guanidine group of Arg357, amide oxygen of Asp170 and alcoholic hydroxyl of the side chain of Asp170 in Im0^{-O2} and Im1; after binding to Mn4 as the new W1, its H-bonds to O4 and Arg357 are broken while others reserve. This indicates the loss of stabilization energy from the H-bond interactions can be fully compensated by W7-Mn4 binding. We hope these can address your concerns. We have added Supplementary Text S6 for this part.

(2) The pathway for W2 unbinding is also unclear. How is the transit of W2 to the bulk? Is there any evidence of the exit of W2 in any S₀ structure?

Response: Thank you for your question for us to clarify. The transfer of W2 to the bulk is, in terms of behavior, very similar to the last step of the proposed mechanism of water exchange for the OEC. According to Fig. 4 in the revised manuscript, a low-barrier route for W2(H₂O) de-coordination and release from Mn4(III) has been located. In the model calculation, the Mn4-W2 distance is changed from 2.18 Å (W2 bound) to 3.99 Å (W2 unbound). A distance of ~ 4 Å is generally considered in the water exchange region, and in any case, to move the water even further out costs very little energy. Due to the size limitation of the model, only a finite value ~ 4 Å of the Mn4-W2 distance can be obtained but this distance adequately illustrates its tendency to transit to the bulk. The situation and explanation are in analogy to substrate water exchange in the S₁, S₂, and S₃ states proposed by Siegbahn.¹² For the S₀ state, there is still no computational studies on the water exchange mechanism, as well as the exit of W2 in any S₀ structure

in literatures, but essentially, they all share the common point that Mn4(III) in a closed-cubane structure serves as the station for water dissociation. Furthermore, in a scheme from the Messinger group based on the experimental data (in their Scheme 3),¹³ W2 unbinding from Mn4 is involved in a closed-cubane structure in their hypothetical water exchange mechanism for the S₀ state. That means W2 departure from Mn4(III) from the cluster makes sense in such circumstances. The Supplementary Text S10 has been added for this.

Minor Suggestions:

1. In Figure 2a, the label "Im2" should be "Im1", and "Im3" should be "Im2".

Thank you. Fixed.

Reviewer #3 (Remarks to the Author):

The manuscript entitled “Closing Kok’s Cycle of Nature’s Water Oxidation Catalysis” by Yu Guo, Lanlan He, Yunxuan Ding, Lars Kloo, Dimitrios A. Pantazis, Johannes Messinger, and Licheng Sun presents a DFT-based theoretical study describing the last events occurring in the catalytic cycle of Photosystem II, subsequent to the molecular oxygen evolution, leading to the restoration of the initial state of the cycle.

The Mn₄Ca cluster placed in the catalytic core of the Photosystem II is responsible for the water splitting reaction in the natural oxygenic photosynthesis, occurring along the five steps (S₀-S₄) of the Kok cycle. In the past years, the events occurring between the S₀ and S₃ states have been extensively characterized experimentally as well as theoretically. Recently, new experiments have been reported (Ref. 9 and 12 in the manuscript) shedding light on the S₃ → S₄ → S₀ transition, in which a peroxide bond between two oxygen atoms is formed, an O₂ molecule is released from the cluster and a water molecule is inserted in the Mn₄Ca cluster to restore the S₀ state. Still, these experiments did not provide information on the mechanism of water insertion and the S₀ reconstitution subsequent to the molecular oxygen evolution. These events have been previously investigated theoretically in different works, suggesting either a stepwise occurrence of oxygen release and water insertion (see e.g. Ref. 20 and 26) or a concerted mechanism (see e.g. Shoji et al. *J. Phys. Chem. B* (2018) 122:6491). In this work, combining ab-initio MD simulations, DFT-based energy calculations on snapshots extracted from the ab-initio MD simulations, and MEP calculations, the mechanism of the S₀ restoration occurring after the oxygen release (assuming the scenario of a stepwise occurrence of oxygen release and water insertion) was explored.

Differently from previous published theoretical works, the results reported in the present manuscript indicate that a water molecule bound to the calcium ion of the cluster (namely W3), is inserted in the cluster, binding the Mn1 ion and therefore originating a closed-cubane structure. In this conformation an additional water molecule (namely W7) binds the Mn4 ion. These two events occur spontaneously within 30 ps of MD simulations, indicating a more or less absence of an energy barrier. Subsequently, a water molecule detaches from the Mn4 ion and the closed-cubane structure is converted into the open-cubane structure by breaking a bond between Mn1 and W3 and forming a bond between Mn4 and W3, thus restoring the (experimentally known) open-cubane structure of the S₀ state. This new proposed mechanism for the S₄ to S₀ transition appears plausible based on the presented calculations and imply the presence of reversible isomerization of the S₀ state, in line with that already suggested for other states of the Kok cycle.

The article appears well written, highly significant and of broad interest. The standard of the calculations carried out in the present work is of high level as can be

expected from these authors and appropriate for well describing the investigated system. However, I have some concerns that should be addressed before recommending the publication of the manuscript on Nature Communications.

Response: Thank you very much for your understanding and favorable comments on our work!

1. The spontaneous insertion in the cluster of the W3 water molecule binding the Mn1 ion as well as the spontaneous binding of the W7 water molecule to Mn4, occurring both in 30 ps of ab initio MD simulations, could be in my opinion artificially facilitated by the absence of the surrounding water molecules and protein residues tightly H-bonded with water molecules present in the simulated system. The system simulated by the authors is relatively large (369 atoms) and they claim that such a dimension is enough to capture "the dynamic phenomena involved in water insertion". I'm not so sure that this is necessarily true. In this sense, an explicit treatment of the environment around the simulated region by means of e.g. QM/MM description, could lead to different results. It could prevent the spontaneous binding of the above-mentioned water molecules associated to the relocation of other water molecules (see e.g. the binding of W7 to Mn4 with the relocation of W8 placed in a water channel truncated in the simulated system). Also, at page 10, "This is not a model artifact..." sounds a bit too strong to me. How can this possibility be totally excluded? Similar mechanisms known to occur in organic metallic chemistry can corroborate the findings of the authors, but not exclude the presence of artifacts. This point should be addressed by the authors, either explaining convincingly why their treatment/model should surely avoid artifacts or pointing out the limits of their calculations.

Response: Thank you for pointing out the possible limitations of our model that we need to clarify and the suggestion to make our language more scientific and accurate. For the first point you mentioned, we have modified 'capture the dynamic phenomena involved in water insertion' to 'representing the conformational changes of the cluster affected by water insertion' in the revised manuscript and the Supplementary Text S7 is specially added for this in the revised supplementary information. It is likely and actually common that different models and methodologies might lead to different results, typically like (pure) QM and QM/MM methods as you indicated. While it is highly flexible to choose either QM or QM/MM for this system, they have both advantages and limitations.^{64,65} QM methods provide a rigorous treatment of electronic structure for specific regions of interest within large systems and provide insights into the detailed mechanisms and energetics of chemical processes; QM calculations can be computationally demanding and compromises the model size which may not fully capture the effects of the surrounding environment, such as solvent or protein interactions. QM/MM methods combine the accuracy of QM calculations with the

efficiency of MM calculations for the surrounding environment, and can capture the effects of the environment more realistically; QM/MM involve approximations and simplifications in the treatment of QM, MM regions, QM/MM boundary and their interactions, which can introduce some level of errors; QM/MM calculations can still be computationally demanding, particularly for large systems (in part due to the difficult self-consistency and convergence between QM and MM regions) and long simulation time and so normally a small QM region is defined. While there have been already a number of excellent publications on the OEC employing QM/MM models^{38,40,53,54,66-71}, for the present study, we chose to use *ab initio* (pure QM description) MD on a large model (369 atoms) based on the following considerations. The chemical event in question is focused on the structural evolution of the central Mn cluster under the interaction of waters in close proximity which involves water binding to Mn/Ca and proton transfer between Mn ligands, so an accurate and intact QM region with as much environment as possible around the Mn cluster should be more important and competent for capturing the events. To our best knowledge, the QM size is by far the largest one among all the molecular dynamic studies on the OEC, and the special GPU acceleration customized in the software employed enables a long simulation time on such a large QM model. Adding the outer MM region based on this could be in principle a plus seen from the integrity of the protein environment, but practically we are not sure about the performance efficiency and accuracy given the complexity mentioned above when it combines with this QM treatment. We note there are already a series of excellent QM/MM MD (Car-Parrinello MD, with much smaller QM region) publications in this field^{14,39,53,54,58} but we adopt here a technically very different methodology (Born-Oppenheimer MD) to simulate cluster evolution. Anyway, worth to try QM/MM MD in future. Regarding the spontaneous movements of W3 binding to Mn1 and W7 binding to Mn4, they are notably observed in our simulations, but they make chemical sense and the simulations have taken into account all the necessary H-bond interactions of the key groups. W3 is H-bonded to W5 and HOH511 which are further H-bonded to W6 which is further H-bonded to Asn181; W7 is H-bonded to O4, W8, guanidine group of Arg357, amide oxygen of Asp170 and alcoholic hydroxyl of the side chain of Asp170; W8 is H-bonded to HOH574, amide oxygens of Asp170 and Asn87. Since all these H-bonded joint groups are explicitly included in the model by QM description, their influences on the movement of W3, W7, and W8 should have been embodied in the MD simulations. Thus it can be seen that even for the most distal waters such as W6 and W8, the H-bond interactions with other surrounding waters or protein residues are also explicitly included. The more distant (approximately >8 Å away from the cluster) water molecules and protein residues that are not included in the model do not have direct H-bond interactions to any of W3, W7 and W8; they may

produce very minor electrostatic and van der Waals interactions that we assume as insignificant and have little impact (especially obstructive effect) on the movement of W3, W7, and W8. The propensity of W3 entering the cavity of the cluster is driven by the largely stabilized structure of the cluster from three unsaturated 5-coordinated Mn(III) (Mn1, Mn3 and Mn4) to saturated 6-coordination. The reason why W3 binding to Mn1 first instead of Mn4 is related to their discrepant charge distributions and the steric effect of the slightly rotated W2 on Mn4, and the H-bond interaction between W2 and W3. W5, which was H-bonded to W3, is pulled by the W3 motion and then occupies the original position of W3 on Ca. W5 is influenced in such a way and is also reflected in literatures on the S₂-S₃ transition^{72,73}. The subsequent W7 binding to Mn4 is a straightforward outcome of the ‘pivot/carousel’-like ligand reorganization around the dangler Mn4, because of the vacant coordination sphere is created toward the free water W7 along with expansion of the \angle Mn3-O4-Mn4 angle by the W3 entrance, W1 translocation to *trans*-O4 and W2 protonation which further promotes W7 binding to Mn4. W8, which was H-bonded to W7, is pulled by W7 motion in the way similar to W5. Thus, the observed phenomena especially related to the movement of waters around Ca and Mn4 are chemically sound.

For the second point you mentioned, we acknowledge that certain events observed in the simulations may arise from the definition of the model, and thus we have revised the text to clarify this point. We also removed the statement that precludes the model-related artifacts and the limits of our calculations have been reflected in Supplementary Text S7. After formation of the pre-S₀(W2-bound) state upon H⁺ release in Im2 (whose rationality as an intermediate has been explained above), the ‘artifacts’ for assigning W2 dissociation as the next step is driven by the progression to the open-cubane S₀ state that has been experimentally identified. Thus it is assumed as a necessary step before the structural isomerization of W3(OH⁻) ligand transfer. Another point regarding the limits of the calculations is about the precision of the reaction energetics of W2 unbinding. Due to the approximations in the accuracy of DFT and the model size limitations by a cluster model^{64,74,75}, errors of the obtained energetics are normally estimated to be within a few kcal/mol, and could be varying among different choices of computational details. Since there are very limited referable benchmarks associated to experimental data for special cases, the significance of these values may be more qualitative than quantitative but does not undermine the validity of the conclusion which can be still reserved within the error bar. Despite the unavoidable model artifacts and errors, we consider the dissociation of W2(H₂O) at this state as a chemically sensible process. The transfer of W2 to the bulk is, in terms of behavior, very similar to the last step of the proposed mechanism of water exchange for the OEC. According to Fig. 4 in the revised manuscript, a low-barrier route for W2(H₂O) de-coordination and

release from Mn4(III) has been located. In the model calculation, the Mn4-W2 distance is changed from 2.18 Å (W2 bound) to 3.99 Å (W2 unbound). A distance of ~ 4 Å is generally considered in the water exchange region, and in any case, to move the water even further out costs very little energy.¹² Due to the size limitation of the model, only a finite value ~ 4 Å of the Mn4-W2 distance can be obtained but this distance adequately illustrates its tendency to transit to the bulk. The situation and explanation are in analogy to substrate water exchange in the S₁, S₂, and S₃ states proposed by Siegbahn.¹² For the S₀ state, there is still no computational studies on the water exchange mechanism, as well as the exit of W2 in any S₀ structure in literatures, but essentially, they all share the common point that Mn4(III) in a closed-cubane structure serves as the station for water dissociation. Furthermore, in a scheme from the Messinger group based on the experimental data (in their Scheme 3), W2 unbinding from Mn4 is involved in a closed-cubane structure in their hypothetical water exchange mechanism for the S₀ state¹³. That means W2 departure from Mn4(III) from the cluster makes sense in such circumstances. This has been demonstrated in Supplementary Text S10.

2. If I've understood correctly, the energies reported in Fig. 2c have been calculated on single snapshots of ab initio MD simulations, I mean single point calculations. If I'm right, I would suggest repeating the calculations on different snapshots (e.g. for Im2 state not just the last snapshot but 10 snapshots taken from the last ps of simulation at interval of 0.1 ps, and similarly for Im0 and Im1) and provide an average value with a standard deviation, being the kind of energy calculation extremely sensitive to the particular geometry chosen for the calculation.

Response: Thank you for your care and suggestion. We apologize for the unclear description and insufficient information on the energetics in the trajectories in the original version. The energies reported in Fig. 2c (in the original version) were not simply single point calculations, but free energies differences based on the fully optimized (with backbone constraints) structures on the snapshots extracted from the three phases for Im0^{-O₂} (first snapshot), Im1 (last snapshot) and Im2 (last snapshot). The first, instead of the last snapshot, is selected for Im0^{-O₂} for sake of better reflecting the shape of the Mn cluster in the initial stage immediately after the release of O₂. The reasons why single point energies were not adopted is as follows. As shown in Supplementary Figs. S6-S7, the energies extracted from the large model used in the BO-AIMD simulations include significant dynamics in weak but plentiful secondary coordination modes involving the protein backbones and explicit water molecules. This causes quite large variation in total energy for the system, and thus statistics in this aspect may be less informative to reliably represent the energy change, even if snapshot averages are used. This is likely to partly propagate even for a smaller model unless

geometrically optimized. Since the architecture and Mn/Ca coordination sphere of the cluster remain unchanged after ca. 15 ps, meaning basically equilibrated water insertion dynamics, a simulation time around 30 ps should be adequate for structural evolution of the cluster. In any case, it is expected that extending the simulation time beyond 30 ps, for instance 100 ps or even longer, is likely to give one of the three outcomes: 1) a nice convergence and a flat $E(\text{tot})$, but this seems unlikely because of the many small changes in secondary coordination structures probably rendering a very slow convergence; 2) a continued slow (i.e. rather flat but still slanting) downhill evolution, for the reasons given in 1); 3) the emergence of a long-period oscillation in $E(\text{tot})$ never really settling to a steady $E(\text{tot})$; this seems the most likely situation for large model systems, just like the case here. For these reasons, besides the extensive computational cost, it is not obvious that very long simulations would solve the problem. Furthermore, the MD simulations capture the inherent thermal fluctuations of the system and the energy output corresponds to instantaneous values at different time points with severe randomness. However, in experimental measurements, energies are typically obtained by averaging over multiple measurements or ensemble sampling, which may be better related to local minima on a potential energy surface. Thus in our opinion, while comparing instantaneous MD energies with experimentally averaged energies may not provide a direct correspondence, more reliable energetics should be extracted from the optimized models of snapshots, as shown in the energies in Fig. 2c (in the original version). Geometry optimizations on the obtained snapshots can eliminate unreasonable or exaggerated energy rise/fall brought by some random, excessive and redundant local structural fluctuations which are common for complex systems during MD simulations, and is probably the most sensible strategy to use. Because geometry optimizations result in local minima on the potential energy surface, using neighboring snapshots with similar structures seems unnecessary. Above all, the purpose for calculating the energetic change from $\text{Im}0^{-\text{O}_2}$ to $\text{Im}1$ to $\text{Im}2$ is to show the thermodynamic feasibility for the structural evolution of the cluster. It is inadequate to build a direct correlation to experimental measurements based on the present model for which other parts and events beyond the local structural evolution surrounding the cluster are not explicitly covered, so there is also no point to pursue an accurate energy difference. This is fortunately inspired by Reviewer 1, who raised the issue about comparison with the energetics of the donor side reactions of PSII. Consequently, in order to avoid unnecessary misunderstandings, we decide to remove the energy profile from the main text but still keep the data in Supplementary Table S1. These have been shown in Supplementary Text S8.

3. The authors used IRC analysis to calculate MEP. I think that, considering the broad audience of the journal, it would be useful to better explain (in the supporting

information) the general scheme behind such a calculation. Advantage and disadvantage (if present) between potential energy surface walking methods and Chain-of-states methods (as NEB, already employed in the investigation of the same mechanism studied here) could be also reported. I would also report in the method section in the main text where the starting geometries for the three reported MEP calculations (Asp61 protonation, water detachment, and closed-open cubane transition) come from.

Response: Thank you for your good suggestion. Providing a more detailed explanation of the general scheme behind the IRC analysis would be beneficial, especially considering the broad audience of the journal. The IRC procedure⁷⁶ is well established (being developed for more than 40 years) with plenty of discussions in literatures. It is a widely used method for calculating the minimum energy pathway (MEP) in chemical reactions, which is defined as the steepest-descent path on the potential energy surface (PES) from the transition states (TS) down towards a local minimum, i.e. reactant and/or product. This can be done, in principle, in any coordinate system. An IRC path is defined similarly but in particular, in mass-weighted coordinates using the Hessian matrix to predict the downhill direction, which means that instead of the steepest descent direction it follows that of the maximum instantaneous acceleration. In the present study, TSs were located by the Berny algorithm⁷⁷ and transit-guided quasi-Newton (STQN) method⁷⁸ embedded in Gaussian. The energy profile is obtained as well as the length and curvature properties of the path, providing the basic quantities for an analysis of the reaction path. Practically, to perform an IRC calculation, several steps are typically involved: 1) the TS geometry is optimized to find the stationary point on the PES; 2) the Hessian matrix is calculated at the TS geometry to obtain the vibrational frequencies and normal modes; 3) the atoms are displaced along the imaginary vibrational mode associated with the lowest frequency to initiate the IRC calculation; 4) the differential equations of motions are numerically integrated to determine the MEP, and integration is typically carried out using algorithms such as the Hessian-based predictor-corrector integrator⁷⁹; 5) the calculated MEP can be visualized using various software tools to gain insights into the reaction mechanism.

Instead of first optimizing a TS by surface walking⁸⁰ and calculating a reaction path afterward, both can be obtained simultaneously from the chain-of-states methods, such as nudged elastic band (NEB)⁸¹, which was developed much later than IRC. NEB optimizes a series of intermediate images positioned between the initial and final states of the reaction. These images are evenly spaced along the reaction path and are optimized to find the lowest energy configurations while maintaining a fixed separation from their neighboring images. Spring forces are introduced between adjacent images to simulate the elastic behavior of the band. These spring forces act to guide the atoms

along the reaction path and prevent excessive distortion. The magnitude of the spring forces is typically set to zero for the initial and final states. NEB calculates the tangent forces acting on each image, which represent the direction of maximum energy change along the reaction path. To ensure that the atoms move along the MEP rather than being influenced solely by the tangent forces, the NEB forces are obtained by projecting out the component of the forces that is perpendicular to the band. By optimizing the positions of the intermediate images while considering the spring forces and projecting the forces, NEB provides a method to explore and determine the MEP and transition states on the potential energy surface.

While IRC is more widely used in the molecular community, NEB seems more popular in the solid state and surface physics/chemistry. IRC and NEB can be competent in their respective fields, but we have some practical considerations. Below are some general advantages and limitations for IRC and NEB. IRC calculations provide a detailed analysis of the reaction pathway along the MEP and allow for the visualization and understanding of the bond-breaking and bond-forming events during a reaction. IRC calculations are conceptually simpler and are generally computationally faster compared to NEB because IRC follows the steepest descent path along the MEP (as long as an accurate Hessian for TS is given initially) without the need for interpolation schemes or optimization of intermediate images. The accuracy of IRC results can be sensitive to the initial guess for the TS geometry. If the initial guess is not close to the true transition state, the IRC path may not accurately represent the reaction pathway, so it is crucial to make reasonable initial guesses to obtain reliable results. In complex systems, there may be multiple possible reaction pathways and transition states. IRC calculations may not effectively explore these alternative reaction pathways, as they are limited to following the steepest descent path from the starting geometry. The main advantage of NEB is that only gradients are required in regular optimizations or surface scans (no exact Hessian required) but unlike surface scans, the method converges to the MEP and allows convenient saddle-point optimization in the same job. Surface scans are also strongly biased towards the choice of the reaction coordinate and can often end up far from the MEP, while NEB has no such bias (there is a small bias towards the initial interpolated path). The accuracy of NEB results can be sensitive to the initial guess for the positions of the intermediate images along the reaction path. If the initial guess is far from the true path, it may require more iterations and adjustments to converge on the MEP accurately. This sensitivity highlights the importance of making reasonable initial guesses for the intermediate states. The choice of interpolation scheme can affect the accuracy and convergence of the NEB calculations. Different interpolation schemes, such as linear or higher-order polynomial interpolation, may yield different results. The selection of an appropriate interpolation scheme is

crucial for obtaining reliable and accurate results. NEB calculations can be computationally expensive, especially for larger and more complex systems. The optimization of multiple images along the reaction path requires additional computational resources, including time and memory. The computational cost may limit the size and complexity of the systems that can be studied using NEB. The above content is added as Supplementary Text S15.

Thank you for reminding us to show more clearly the details of the MEP calculations. The starting geometry for the MEP calculation for Asp61 protonation comes from the last snapshot of the BO-AIMD simulation. Considering the feasibility and efficiency in massive Hessian calculations, the snapshot was truncated to consist of 223 atoms for MEP, without losing possible influence from surrounding necessary groups on Asp61 protonation (as well as the following water detachment and closed-open cubane transition). For the water detachment, the starting geometry for MEP was based on the product state of Asp61 protonation by removal of the protonated Asp61 (for its reported high flexibility and rotation after protonation, shown in Supplementary Text S13). For the closed-open cubane transition, the starting geometry for MEP was based on the product state of W2 dissociation by further removal of W2 (assumed to release to the bulk). These are added in the Methods section.

Reviewer #4 (Remarks to the Author):

The manuscript “Closing Kok’s Cycle of Nature’s Water Oxidation Catalysis” calculated the reset process of $\text{Mn}_4\text{CaO}_{5(6)}$ cluster during the final $\text{S}_3 \rightarrow (\text{S}_4) \rightarrow \text{S}_0$ transition with molecular dynamics simulations combined with density functional calculations. The results suggest a likely missing link for closing the Kok’s cycle. The structural isomerism in the S_0 state is proposed theoretically. The topics are crucial in understanding the nature’s water oxidation catalysis, and the results are very interesting and inspiring for the design of artificial water oxidation catalysts. I recommend the publication of the present manuscript in Nature Communications with the following issues well addressed.

Response: Thank you very much for your understanding and positive comment on our work!

1. Although the structure of S_4 is claimed to be $\text{Mn(IV)-O}\bullet$ radical, the peroxide intermediate formed by oxo-oxyl coupling has not yet been confirmed experimentally. The different structure of the peroxide intermediate may form different structure after releasing O_2 , which may influence the following reset process of $\text{Mn}_4\text{CaO}_{5(6)}$ cluster. This should be considered and added.

Response: Thank you for your suggestion. You are correct that the peroxide intermediate formed by oxo-oxyl coupling has not yet been confirmed experimentally. Bhowmick et al. did not capture the peroxide intermediate (typical O-O distance 1.4~1.5 Å) by XFEL crystallography for the timepoints during the S_3 - S_0 transition, and their indication for the peroxide-like species is mostly suggested by the timescale and kinetics for the observed structural changes of the cluster (there is a delay between the onset of O-O bond formation and the decrease of the Ox electron density). The evidence from Greife et al.⁹ for oxo-oxyl coupling is to some degree embodied from the computational results. It is also true that different peroxide intermediates may form different structures after O_2 release and influence the following resetting process. It can be seen that up to now uncertainties still remain in this process which are unable to be figured out by experimental techniques at least for now. This highlights the important role of computational studies as an auxiliary tool to look into the section that is unreachable by experiments, as what we are dedicating to in this study. Aware of the existing uncertainties, we have to make careful deliberation and select the most likely scenario as the foundation of our study, according to the majority of available evidence (while doing all is not possible in a single study). Specifically here, we are confident that we have grasped the most likely mechanism for O-O bond formation as the base for the following Mn_4CaO_5 resetting process. So far, the preponderance of available evidence from most experimental and theoretical studies, including XFEL crystallography^{8,28}, time-resolved spectroscopies⁹, isotope labeling water exchange

experiments²⁹, DFT modelling^{15,32-35} and large-scale QM/MM simulations^{19,36-39}, etc., strongly supports the oxo-oxyl coupling mechanism. Also, both of the two recent breakthroughs by Bhowmick et al.⁸ and Greife et al.⁹ (as the background of our present study) have recommended it as the most viable or even compelling pathway, respectively, from the perspectives of structural intermediates by XFEL crystallography and time-resolved microsecond Fourier transform infrared spectroscopy (FTIR) combined with computational chemistry, respectively. *Actually, as long as the substrates are O5 and Ox, regardless of their coupling way (in terms of electronic configuration of Ox and open/closed conformation of the cluster^{5,31,82,83}) or the variable spatial orientation of the peroxide moiety^{37,38}, the resetting process suggested here is generally applicable because the Mn₄CaO₄ cluster with the structural cavity left by release of O₂ (O5-Ox) is set as the starting point.* However, our work does not aim to rule out or decrease the significance of other possibilities of O-O bond formation^{68,70,84,85} that are worth consideration in parallel. The corresponding cluster resetting process in these scenarios would be totally different, since the structural cavity within the cluster left by O₂ release would not be the same, depending on the position of substrates. They should be studied specifically as well if compelling evidence against the current O5-Ox coupling would appear in future. One should be open-minded about this, but for the present study, we must follow currently the most convincing mechanism of O₂ formation for exploring the subsequent possible routes toward the S₀ state. These have been clarified in Supplementary Text S1.

2. As calculated in Greife et al., the single-electron—multi-proton step facilitates the critical step of Mn(IV)-O● formation, which relies on a specific location of water molecules and hydrogen-bond interaction. In principle, the specific location of water molecules and hydrogen-bond interaction should play more crucial roles in the reset process of Mn₄CaO₅ cluster. This should be studied and discussed in detail.

Response: Thank you for raising this important point. The single-electron—multi-proton transfer event proposed by Greife et al.⁹ includes single electron transfer from Ox to Y_Z●, concerted with multiple proton transfer from Ox(OH⁻) to W3(H₂O), W3 to W2(OH⁻) and W1(H₂O) to Asp61(COO⁻). It is the slowest step in photosynthetic O₂ formation with a moderate energetic barrier and marked entropic slowdown, leading to the oxygen-radical S₄ state for fast O-O bonding and O₂ release. Obviously, the specific location of water molecules and hydrogen-bond (HB) interactions between them play a very important role for facilitating the critical step of Mn(IV)-O● formation. The multi-proton transfer would not take place without the water molecules in HB network located along the route. Greife et al. also tested the cases of electron transfer from Ox without proton movements, but this would highly destabilize the system and this indicates the importance of simultaneous proton transfer in the HB network. The HB interactions of

Ox-W3 (formation after Ox-Glu189 breakage), W3-W2, and W1-Asp61 are essentially the effective ones. Since the free energy barrier from the $S_3Y_Z\bullet$ to S_4 transition was determined to be 13.6 kcal/mol as the kinetic bottleneck, the concerted proton transfer cannot be captured in a QM/MM MD simulation in tens of picoseconds, and minimum energy path (MEP) calculations (by the NEB method) were employed to study the proton transfer and determine the barrier (accompanied by electron transfer).

For our present study, the specific locations of water molecules and HB interactions are also crucial for the resetting process of Mn_4CaO_5 cluster. Specifically, Our simulations show the crucial roles of the water molecules at least closely around the cluster, i.e. W1, W2(OH⁻), W3, W5, W6, W7 and W8 and the HB interactions of W5-W6, W3-W5, W3-W2 and W7-W8; Due to limitation of the model size, HB interactions between the waters locating further away (out of the model, with minor effect for the cluster resetting process) cannot be reflected but their possible motions driven by the HB interactions can also be expected. Among them, here we emphasize the particularly crucial role of the Ca-bound W3 and the Mn4-bound W2 and the HB interaction between them in the cluster resetting process. During the process of W3 moving to the cavity, it forms strong HB interaction with W2 which is the partial reason (the other reason being Mn charge distribution) for W3 binding to Mn1 instead of Mn4, i.e. binding to Mn1 has more favorable HB orientation to W2 and W2 rotates around Mn4 toward W3 which to some degree hinders W3 approach to Mn4. After Im1 formation, the W3-W2 HB interaction appears more important because the short and strong HB interaction leads to low-barrier (or even barrierless) proton transfer between them which facilitates the following structural evolution. W3 spontaneously deprotonates to W2 which causes W3 approaching to Mn3 (creating the closed-cubane) and the ‘pivot/carousel’-like ligand reorganization around the dangler Mn4, together with W2 rotation after its protonation. In the meanwhile, the cluster is expanded embodied from the elongated Mn1-Mn4 distance and increased $\angle Mn3-O4-Mn4$, and the resultant geometric change of the Mn4 coordination has made an empty coordination layer toward W7 from the O4 channel. W7 binding to Mn4 occurs immediately after (or almost synchronous with) W2 protonation because of further rotations of W2 and W1 and decreased structural *trans* effect on Mn4. The HB interaction of W3-W5 makes W5 occupy the original coordination of W3 on Ca. Besides, W6 and W8 motions pulled by the HB interactions of W5-W6 and W7-W8 (and similar others out of the model) are indispensable for recovery of the surrounding water environment of the OEC. Different from the single-electron multi-proton transfer with a mediate barrier for Greife et al., the present case does not involve electron transfer because there is no electron hole on Y_Z or any radical on the ligands; the present case only involve one proton transfer from W3(H₂O) to W2(OH⁻), which is considered barrierless (or almost) since it is observed

spontaneously to take place in the picosecond timescale (we surmise the O_x-H deprotonation in Greife et al.'s case should be more demanding than H₂O deprotonation). This is also consistent with the fast kinetics for the S₄-S₀ transition post to O-O bond formation. The Supplementary Text S17 has been added for this part.

Besides, we also emphasize that the events we have simulated do not cover the whole process of the S₀ restoration, which, apart from the O₅ recovery, may include kinds of structural rearrangements (possibly caused by proton release) in the protein environment far away from the Mn₄CaO₅ cluster which cannot be considered in our model and simulations. Bhowmick et al.⁸ observed that from 1,200 to 4,000 μs several structural changes occur, and most of these changes are indicative of O₂ release and/or water insertion. This indicates that O₂ release and refilling of the cluster by bulk water and resetting of the catalytic center occur over an extended timescale. Additional distances changes between Y_Z and His190 are observed between 2,000 and 4,000 μs, which may be due to rearrangement of the HB network related to the last proton release. In our proposal, H⁺ is released from W1 (as the new W2) to the lumen *via* Asp61 in Im2, and this is assumed to be the second proton release during the S₃-S₀ transition. After gated by Asp61, the released H⁺ would pass by substantial water and protein residue groups located in the proton channel of PSII which may cause pronounced variations in rearrangement of the HB network (including water-water, protein-protein, and water-protein interactions) coupled to protein dynamics²⁶ and possibly other unknown structural changes, together responsible for the extended timescale between 2,000 and 4,000 μs. For example, the increase of the Y_Z-D1-H190, Ca-D1-E189 and Mn4-O5 distances and decrease of the Mn1-Mn4 distance were observed. These have been clarified concisely in the Discussion section in the main text and in detail in Supplementary Text S16.

3. As studied in Bhowmick et al., the structural changes from 1,200 to 4,000 μs indicates that O₂ release and refilling of the cluster by bulk water and resetting of the catalytic center occur over an extended timescale. But this manuscript suggests that the resetting of the catalytic center is at picosecond timescale. Why does this huge difference exist? If the ultrafast resetting indeed completes at picosecond timescale, what happens from 1,200 to 4,000 μs?

Response: Thank you for this important question and we should make it clear enough. Bhowmick et al.⁸ observed that from 1,200 to 4,000 μs, several structural changes occur, and most of these changes are indicative of O₂ release and/or water insertion. This indicates that O₂ release and refilling of the cluster by bulk water and resetting of the catalytic center occur over an extended timescale. Within this period, 1,200 and 2,000 μs are the two essential timepoints that are closely related to our work. According to their identification, the 1,200 μs snapshot signifies the onset of O₂ evolution; the 2,000

μs snapshot, without Ox on the electron density omit map, indicates completion of binding of a water that refills the vacant site formed by O_2 release. On this basis, the process we suggest in the present study should in principle transiently reside between these two timepoints, but being a very short period seen from the picosecond timescale for water insertion and nanosecond timescale for the subsequent closed-to-open cubane transformation of the cluster. It is understandable that the ultrafast conversion cannot be captured by the XFEL crystallography with hundreds of microseconds as the interval. According to Bhowmick et al., additional distances changes between Y_Z and His190 are observed between 2,000 and 4,000 μs , which may be due to rearrangement of the HB network related to the last proton release but are not well understood currently. In our proposal, H^+ is released from W1 (as the new W2) to lumen *via* Asp61 in Im2, and this is assumed to be the second proton release during the $\text{S}_3 \rightarrow \text{S}_0$ transition. After gated by Asp61, the released H^+ would pass by substantial water and protein residue groups located in the proton channel of PSII which can cause rearrangement of the HB network and other structural changes responsible for the extended timescale between 2,000 and 4,000 μs . For example, the increase of the Y_Z -D1-H190, Ca-D1-E189 and Mn4-O5 distances and decrease of the Mn1-Mn4 distance were observed. Thus for your concern, we fully respect and acknowledge the timescale observed by Bhowmick et al., and clarify that neither we suggest the resetting of the catalytic center is at picosecond timescale nor our suggested process covers the whole period from 1,200 to 4,000 μs ; but rather the mechanism/progression revealed by our MD simulations and MEP calculations represents only an ultrashort phase (from picosecond to nanosecond timescale) embedded in the timepoints between 1,200 (precisely some timepoint after this when O_2 has been released) and 2,000 μs . Therefore, our suggestion does not conflict with Bhowmick et al.'s observation. Besides, it is noted that in Bhowmick et al.'s definition, the final S_0 state is assigned to 3F(200 ms); however, by the 2,000 μs timepoint after 3F, the O5 omit map density is restored considerably compared with the S_3 and S_0 states, which indicates water insertion after O_2 release has occurred, although not yet the final S_0 state with fully restored electron densities and other indicators. In our modelling and definition, the final S_0 state (i.e. the S_0^A state) is reached as long as the open-cubane structure is formed within the Mn_4CaO_5 cluster, which should correspond to a timepoint close to 2,000 μs . So there is a discrepancy on the definition of the S_0 state between Bhowmick et al.'s and ours. Since our model and simulations cannot and do not aim to represent other structural changes after 2,000 μs toward the $\text{S}_0(3\text{F}(200 \text{ ms}))$ state, semantic misunderstanding might be caused when compared to the experimental findings regarding the extended timescale. Anyway, we have clarified this and hope you can understand. The Supplementary Text S156 has been added for this part.

References

1. Ibrahim, M., *et al.* Untangling the sequence of events during the $S_2 \rightarrow S_3$ transition in photosystem II and implications for the water oxidation mechanism. *Proc. Natl. Acad. Sci. U.S.A.* **117**, 12624-12635 (2020).
2. Zaharieva, I. & Dau, H. Energetics and kinetics of S-State transitions monitored by delayed chlorophyll fluorescence. *Front. Plant Sci.* **10**, 386 (2019).
3. Klauss, A., Haumann, M. & Dau, H. Seven steps of alternating electron and proton transfer in photosystem II water oxidation traced by time-resolved photothermal beam deflection at improved sensitivity. *J. Phys. Chem. B* **119**, 2677-2689 (2015).
4. Nilsson, H., Cournac, L., Rappaport, F., Messinger, J. & Lavergne, J. Estimation of the driving force for dioxygen formation in photosynthesis. *Biochim. Biophys. Acta, Bioenerg.* **1857**, 23-33 (2016).
5. Li, X. & Siegbahn, P.E.M. Alternative mechanisms for O_2 release and O-O bond formation in the oxygen evolving complex of photosystem II. *Phys. Chem. Chem. Phys.* **17**, 12168-12174 (2015).
6. Siegbahn, P.E.M. Water oxidation in photosystem II: oxygen release, proton release and the effect of chloride. *Dalton Trans.*, 10063-10068 (2009).
7. Marcus, Y. Thermodynamic functions of transfer of single ions from water to nonaqueous and mixed solvents: Part 2 - Enthalpies and entropies of transfer to nonaqueous solvents. *Pure Appl. Chem.* **57**, 1103-1128 (1985).
8. Bhowmick, A., *et al.* Structural evidence for intermediates during O_2 formation in photosystem II. *Nature* **617**, 629-636 (2023).
9. Greife, P., *et al.* The electron-proton bottleneck of photosynthetic oxygen evolution. *Nature* **617**, 623-628 (2023).
10. (!!! INVALID CITATION !!! 1,2).
11. Schwiedrzik, L., Rajkovic, T. & González, L. Regeneration and degradation in a biomimetic polyoxometalate water oxidation catalyst. *ACS Catal.* **13**, 3007-3019 (2023).
12. Siegbahn, P.E.M. Substrate water exchange for the oxygen evolving complex in PSII in the S_1 , S_2 , and S_3 states. *J. Am. Chem. Soc.* **135**, 9442-9449 (2013).
13. de Lichtenberg, C. & Messinger, J. Substrate water exchange in the S_2 state of photosystem II is dependent on the conformation of the Mn_4Ca cluster. *Phys. Chem. Chem. Phys.* **22**, 12894-12908 (2020).
14. Capone, M., Narzi, D. & Guidoni, L. Mechanism of oxygen evolution and Mn_4CaO_5 cluster restoration in the natural water-oxidizing catalyst. *Biochemistry* **60**, 2341-2348 (2021).
15. Siegbahn, P.E.M. Water oxidation mechanism in photosystem II, including oxidations, proton release pathways, O-O bond formation and O_2 release. *Biochim. Biophys. Acta, Bioenerg.* **1827**, 1003-1019 (2013).
16. Krewald, V., *et al.* Spin state as a marker for the structural evolution of nature's water-

- splitting catalyst. *Inorg. Chem.* **55**, 488-501 (2016).
17. Krewald, V., *et al.* Metal oxidation states in biological water splitting. *Chem. Sci.* **6**, 1676-1695 (2015).
 18. Ames, W., *et al.* Theoretical evaluation of structural models of the S₂ state in the oxygen evolving complex of photosystem II: protonation states and magnetic interactions. *J. Am. Chem. Soc.* **133**, 19743-19757 (2011).
 19. Allgöwer, F., Gamiz-Hernandez, A.P., Rutherford, A.W. & Kaila, V.R.I. Molecular principles of redox-coupled protonation dynamics in photosystem II. *J. Am. Chem. Soc.* **144**, 7171–7180 (2022).
 20. Shevela, D., Kern, J.F., Govindjee, G. & Messinger, J. Solar energy conversion by photosystem II: principles and structures. *Photosynth. Res.* **156**, 279–307 (2023).
 21. Debus, R.J. Evidence from FTIR difference spectroscopy that D1-Asp61 influences the water reactions of the oxygen-evolving Mn₄CaO₅ cluster of photosystem II. *Biochemistry* **53**, 2941-2955 (2014).
 22. Shimada, Y., Sugiyama, A., Nagao, R. & Noguchi, T. Role of D1-Glu65 in Proton Transfer during Photosynthetic Water Oxidation in Photosystem II. *J. Phys. Chem. B* (2022).
 23. Hussein, R., *et al.* Evolutionary diversity of proton and water channels on the oxidizing side of photosystem II and their relevance to function. *Photosynth. Res.* (2023).
 24. Yang, K.R., Lakshmi, K.V., Brudvig, G.W. & Batista, V.S. Is deprotonation of the oxygen-evolving complex of photosystem II during the S₁ → S₂ transition suppressed by proton quantum delocalization? *J. Am. Chem. Soc.* **143**, 8324-8332 (2021).
 25. Hussein, R., *et al.* Structural dynamics in the water and proton channels of photosystem II during the S₂ to S₃ transition. *Nat. Commun.* **12**, 6531 (2021).
 26. Guerra, F., Siemers, M., Mielack, C. & Bondar, A.-N. Dynamics of long-distance hydrogen-bond networks in photosystem II. *J. Phys. Chem. B* **122**, 4625-4641 (2018).
 27. Siegbahn, P.E.M. The effect of backbone constraints: the case of water oxidation by the oxygen-evolving complex in PSII. *ChemPhysChem* **12**, 3274-3280 (2011).
 28. Suga, M., *et al.* An oxyl/oxo mechanism for oxygen-oxygen coupling in PSII revealed by an x-ray free-electron laser. *Science* **366**, 334-338 (2019).
 29. de Lichtenberg, C., Kim, C.J., Chernev, P., Debus, R.J. & Messinger, J. The exchange of the fast substrate water in the S₂ state of photosystem II is limited by diffusion of bulk water through channels – implications for the water oxidation mechanism. *Chem. Sci.* **12**, 12763-12775 (2021).
 30. de Lichtenberg, C., *et al.* Assignment of the slowly exchanging substrate water of nature's water-splitting cofactor. *Proc. Natl. Acad. Sci.* **121**, e2319374121 (2024).
 31. Cox, N. & Messinger, J. Reflections on substrate water and dioxygen formation. *Biochim. Biophys. Acta, Bioenerg.* **1827**, 1020-1030 (2013).
 32. Siegbahn, P.E.M. Structures and energetics for O₂ formation in photosystem II. *Acc. Chem.*

- Res.* **42**, 1871-1880 (2009).
33. Guo, Y., *et al.* The open-cubane oxo-oxyl coupling mechanism dominates photosynthetic oxygen evolution: a comprehensive DFT investigation on O-O bond formation in the S₄ state. *Phys. Chem. Chem. Phys.* **19**, 13909-13923 (2017).
 34. Rummel, F. & O'Malley, P.J. How nature makes O₂: an electronic level mechanism for water oxidation in photosynthesis. *J. Phys. Chem. B* **126**, 8214-8221 (2022).
 35. Corry, T.A. & O'Malley, P.J. Electronic-level View of O-O bond formation in nature's water oxidizing complex. *J. Phys. Chem. Lett.* **11**, 4221-4225 (2020).
 36. Song, X. & Wang, B. O-O bond formation and oxygen release in photosystem II are enhanced by spin-exchange and synergetic coordination interactions. *J. Chem. Theory Comput.* **19**, 2684-2696 (2023).
 37. Shoji, M., Isobe, H., Shigeta, Y., Nakajima, T. & Yamaguchi, K. Nonadiabatic one-electron transfer mechanism for the O-O bond formation in the oxygen-evolving complex of photosystem II. *Chem. Phys. Lett.* **698**, 138-146 (2018).
 38. Capone, M., Guidoni, L. & Narzi, D. Structural and dynamical characterization of the S₄ state of the Kok-Joliot's cycle by means of QM/MM molecular dynamics simulations. *Chem. Phys. Lett.* **742**, 137111 (2020).
 39. Narzi, D., Capone, M., Bovi, D. & Guidoni, L. Evolution from S₃ to S₄ states of the oxygen-evolving complex in photosystem II monitored by quantum mechanics/molecular mechanics (QM/MM) dynamics. *Chem. Eur. J.* **24**, 10820-10828 (2018).
 40. Shoji, M., Isobe, H., Shigeta, Y., Nakajima, T. & Yamaguchi, K. Concerted mechanism of water insertion and O₂ release during the S₄ to S₀ transition of the oxygen-evolving complex in photosystem II. *J. Phys. Chem. B* **122**, 6491-6502 (2018).
 41. Isobe, H., *et al.* Theoretical illumination of water-inserted structures of the CaMn₄O₅ cluster in the S₂ and S₃ states of oxygen-evolving complex of photosystem II: full geometry optimizations by B3LYP hybrid density functional. *Dalton Trans.* **41**, 13727-13740 (2012).
 42. Saito, K., Nakagawa, M. & Ishikita, H. pK_a of the ligand water molecules in the oxygen-evolving Mn₄CaO₅ cluster in photosystem II. *Commun. Chem.* **3**, 89 (2020).
 43. Kusunoki, M. S₁-state Mn₄Ca complex of Photosystem II exists in equilibrium between the two most-stable isomeric substates: XRD and EXAFS evidence. *J. Photochem. Photobiol., B* **104**, 100-110 (2011).
 44. Zahariou, G., Ioannidis, N., Sanakis, Y. & Pantazis, D.A. Arrested substrate binding resolves catalytic intermediates in higher-plant water oxidation. *Angew. Chem. Int. Ed.* **60**, 3156-3162 (2021).
 45. Drosou, M., Zahariou, G. & Pantazis, D.A. Orientational Jahn-Teller isomerism in the dark-stable state of nature's water oxidase. *Angew. Chem. Int. Ed.* **60**, 13493-13499 (2021).
 46. Lohmiller, T., *et al.* The first state in the catalytic cycle of the water-oxidizing enzyme: identification of a water-derived μ -hydroxo bridge. *J. Am. Chem. Soc.* **139**, 14412-14424

- (2017).
47. Retegan, M., *et al.* A five-coordinate Mn(IV) intermediate in biological water oxidation: spectroscopic signature and a pivot mechanism for water binding. *Chem. Sci.* **7**, 72-84 (2016).
 48. Chrysina, M., *et al.* Five-coordinate Mn^{IV} intermediate in the activation of nature's water splitting cofactor. *Proc. Natl. Acad. Sci. U.S.A.* **116**, 16841-16846 (2019).
 49. Cox, N., *et al.* Electronic structure of the oxygen-evolving complex in photosystem II prior to O-O bond formation. *Science* **345**, 804-808 (2014).
 50. Guo, Y., Zhang, B., Kloo, L. & Sun, L. Necessity of structural rearrangements for O-O bond formation between O5 and W2 in photosystem II. *J. Energy Chem.* **57**, 436-442 (2021).
 51. Guo, Y., *et al.* How does ammonia bind to the oxygen-evolving complex in the S₂ state of photosynthetic water oxidation? Theoretical support and implications for the W1 substitution mechanism. *Phys. Chem. Chem. Phys.* **18**, 31551-31565 (2016).
 52. Capone, M., Bovi, D., Narzi, D. & Guidoni, L. Reorganization of substrate waters between the closed and open cubane conformers during the S₂ to S₃ transition in the oxygen evolving complex. *Biochemistry* **54**, 6439-6442 (2015).
 53. Narzi, D., Bovi, D. & Guidoni, L. Pathway for Mn-cluster oxidation by tyrosine-Z in the S₂ state of photosystem II. *Proc. Natl. Acad. Sci. U.S.A.* **111**, 8723-8728 (2014).
 54. Bovi, D., Narzi, D. & Guidoni, L. The S₂ state of the oxygen-evolving complex of photosystem II explored by QM/MM dynamics: spin surfaces and metastable states suggest a reaction path towards the S₃ state. *Angew. Chem. Int. Ed.* **52**, 1-6 (2013).
 55. Wang, J., Askerka, M., Brudvig, G.W. & Batista, V.S. Crystallographic data support the carousel mechanism of water supply to the oxygen-evolving complex of photosystem II. *ACS Energy Lett.* **2**, 2299-2306 (2017).
 56. Askerka, M., Wang, J., Vinyard, D.J., Brudvig, G.W. & Batista, V.S. S₃ state of the O₂-evolving complex of photosystem II: insights from QM/MM, EXAFS, and femtosecond X-ray diffraction. *Biochemistry* **55**, 981-984 (2016).
 57. Askerka, M., Vinyard, D.J., Brudvig, G.W. & Batista, V.S. NH₃ binding to the S₂ state of the O₂-evolving complex of photosystem II: analogue to H₂O binding during the S₂→S₃ transition. *Biochemistry* **54**, 5783-5786 (2015).
 58. Capone, M., Narzi, D., Bovi, D. & Guidoni, L. Mechanism of water delivery to the active site of photosystem II along the S₂ to S₃ transition. *J. Phys. Chem. Lett.* **7**, 592-596 (2016).
 59. Coe, B.J. & Glenwright, S.J. Trans-effects in octahedral transition metal complexes. *Coord. Chem. Rev.* **203**, 5-80 (2000).
 60. Quagliano, J.V. & Schubert, L.E.O. The trans effect in complex inorganic compounds. *Chem. Rev.* **50**, 201-260 (1952).
 61. Burdett, J.K. & Albright, T.A. Trans influence and mutual influence of ligands coordinated to a central atom. *Inorg. Chem.* **18**, 2112-2120 (1979).

62. Shustorovich, E.M., Porai-Koshits, M.A. & Buslaev, Y.A. The mutual influence of ligands in transition metal coordination compounds with multiple metal-ligand bonds. *Coord. Chem. Rev.* **17**, 1-98 (1975).
63. Rivalta, I., *et al.* Structural-functional role of chloride in photosystem II. *Biochemistry* **50**, 6312-6315 (2011).
64. Blomberg, M.R.A., Borowski, T., Himo, F., Liao, R.-Z. & Siegbahn, P.E.M. Quantum chemical studies of mechanisms for metalloenzymes. *Chem. Rev.* **114**, 3601-3658 (2014).
65. Ahmadi, S., *et al.* Multiscale modeling of enzymes: QM-cluster, QM/MM, and QM/MM/MD: A tutorial review. *Int J Quantum Chem.* **118**, e25558 (2018).
66. Yamaguchi, K., *et al.* Geometric, electronic and spin structures of the CaMn₄O₅ catalyst for water oxidation in oxygen-evolving photosystem II. Interplay between experiments and theoretical computations. *Coord. Chem. Rev.* **471**, 214742 (2022).
67. Shoji, M., Isobe, H. & Yamaguchi, K. QM/MM study of the S₂ to S₃ transition reaction in the oxygen-evolving complex of photosystem II. *Chem. Phys. Lett.* **636**, 172-179 (2015).
68. Sproviero, E.M., Gascón, J.A., McEvoy, J.P., Brudvig, G.W. & Batista, V.S. Quantum mechanics/molecular mechanics study of the catalytic cycle of water splitting in photosystem II. *J. Am. Chem. Soc.* **130**, 3428-3442 (2008).
69. Sproviero, E.M., Gascón, J.A., McEvoy, J.P., Brudvig, G.W. & Batista, V.S. QM/MM models of the O₂-evolving complex of photosystem II. *J. Chem. Theory Comput.* **2**, 1119-1134 (2006).
70. Kawashima, K., Takaoka, T., Kimura, H., Saito, K. & Ishikita, H. O₂ evolution and recovery of the water-oxidizing enzyme. *Nat. Commun.* **9**, 1247 (2018).
71. Saito, K., Rutherford, A.W. & Ishikita, H. Energetics of proton release on the first oxidation step in the water-oxidizing enzyme. *Nat. Commun.* **6**, 8488 (2015).
72. Kim, C.J. & Debus, R.J. Evidence from FTIR difference spectroscopy that a substrate H₂O molecule for O₂ formation in photosystem II is provided by the Ca ion of the catalytic Mn₄CaO₅ cluster. *Biochemistry* **56**, 2558-2570 (2017).
73. Kim, C.J. & Debus, R.J. One of the substrate waters for O₂ formation in photosystem II is provided by the water-splitting Mn₄CaO₅ cluster's Ca²⁺ ion. *Biochemistry* **58**, 3185-3192 (2019).
74. Siegbahn, P.E.M. The performance of hybrid DFT for mechanisms involving transition metal complexes in enzymes. *J. Biol. Inorg. Chem.* **11**, 695-701 (2006).
75. Cramer, C.J. & Truhlar, D.G. Density functional theory for transition metals and transition metal chemistry. *Phys. Chem. Chem. Phys.* **11**, 10757-10816 (2009).
76. Fukui, K. The path of chemical-reactions-The IRC approach. *Acc. Chem. Res.* **14**, 363-368 (1981).
77. Schlegel, H.B. Optimization of equilibrium geometries and transition structures. *J. Comp. Chem.* **3**, 214-218 (1982).

78. Peng, C., Ayala, P.Y., Schlegel, H.B. & Frisch, M.J. Using redundant internal coordinates to optimize equilibrium geometries and transition states. *J. Comp. Chem.* **17**, 49-56 (1996).
79. Hratchian, H.P. & Schlegel, H.B. Accurate reaction paths using a Hessian based predictor–corrector integrator. *J. Chem. Phys.* **120**, 9918-9924 (2004).
80. Simons, J., Joergensen, P., Taylor, H. & Ozment, J. Walking on potential energy surfaces. *J. Phys. Chem.* **87**, 2745-2753 (1983).
81. Henkelman, G., Uberuaga, B.P. & Jónsson, H. A climbing image nudged elastic band method for finding saddle points and minimum energy paths. *J. Phys. Chem.* **113**, 9901-9904 (2000).
82. Krewald, V., Neese, F. & Pantazis, D.A. Implications of structural heterogeneity for the electronic structure of the final oxygen-evolving intermediate in photosystem II. *J. Inorg. Biochem.* **199**, 110797 (2019).
83. Guo, Y., Messinger, J., Kloo, L. & Sun, L. Alternative mechanism for O₂ formation in natural photosynthesis via nucleophilic oxo–oxo coupling. *J. Am. Chem. Soc.* **145**, 4129–4141 (2023).
84. Zhang, B. & Sun, L. Why nature chose the Mn₄CaO₅ cluster as water-splitting catalyst in photosystem II: a new hypothesis for the mechanism of O-O bond formation. *Dalton Trans.* **47**, 14381-14387 (2018).
85. Shen, J.-R. The structure of photosystem II and the mechanism of water oxidation in photosynthesis. *Annu. Rev. Plant Biol.* **66**, 23-48 (2015).

REVIEWER COMMENTS

Reviewer #1 (Remarks to the Author):

The authors have addressed my concern appropriately. Publication of the revised manuscript is strongly recommended.

Reviewer #2 (Remarks to the Author):

The authors have answered all of the questions. However, there are a few issues that remain to be explained:

(1) The rationale for W1 and W2 not being H₂O at S1 state (and S2 state, with W1 sharing a proton with D61) seems to rely on a 2011 JACS paper (<https://pubs.acs.org/doi/10.1021/ja2041805>) and a 2015 Chemical Science paper by Pantazis (<https://pubs.rsc.org/en/content/articlelanding/2015/sc/c4sc03720k>). In the 2015 paper, they are saying one extra proton would make S2 a high spin state since one extra proton results in J34 of -2.6, compared to -7.6 for one fewer proton, Figure 4 & Table S1). However, it is unclear whether those DFT calculations can properly resolve the relative spin states. A comparative analysis of other protonation states would have provided a more valuable analysis.

It is important to note that the authors indicated in their response that they cannot rule out the possibility of an extra proton. I recommend the authors should indicate that observation in the main text and include a reference to earlier work where that possibility has already been supported by direct comparisons to experimental data.

(2) Regarding W1 breaking the hydrogen bond with D61 to rotate into the position of W2, they do emphasize that this happens only within 1-2 ps in their MD. I guess Mn4 being a Mn³⁺ makes it possible to rotate W1, compared to Mn4 being Mn⁴⁺ with a quantum proton in between W1 and D61. However, it is unclear whether that rearrangement is induced by the specific preparation of the initial conditions. I recommend the authors should analyze how much of that rearrangement depends on the initial conditions of their simulations.

(3) The authors have estimated how much energy is required to detach W2. However, they have not estimated the error of that estimated energy. I recommend they should provide an estimate of the error of their calculation.

Reviewer #3 (Remarks to the Author):

Authors have convincingly addressed all the critical points I had indicated. Therefore, I suggest the publication of the manuscript in its current form in the Nature Communications journal.

I only make one note: in their response to the referees (see, for example, referee 1), the authors state that the work by Guidoni and coworkers is based on QM/MM MD Car-Parrinello. In reality, the works in question were carried out based on QM/MM MD Born-Oppenheimer as implemented in the CP2K package.

Reviewer #4 (Remarks to the Author):

The revised version is acceptable to me.

Reviewer #1 (Remarks to the Author):

The authors have addressed my concern appropriately. Publication of the revised manuscript is strongly recommended.

Response: Thank you again for your nice review!

Reviewer #2 (Remarks to the Author):

The authors have answered all of the questions. However, there are a few issues that remain to be explained.

(1) The rationale for W1 and W2 not being H₂O at S₁ state (and S₂ state, with W1 sharing a proton with D61) seems to rely on a 2011 JACS paper (<https://pubs.acs.org/doi/10.1021/ja2041805>) and a 2015 Chemical Science paper by Pantazis (<https://pubs.rsc.org/en/content/articlelanding/2015/sc/c4sc03720k>). In the 2015 paper, they are saying one extra proton would make S₂ a high spin state since one extra proton results in J₃₄ of -2.6, compared to -7.6 for one fewer proton, Figure 4 & Table S1). However, it is unclear whether those DFT calculations can properly resolve the relative spin states. A comparative analysis of other protonation states would have provided a more valuable analysis.

It is important to note that the authors indicated in their response that they cannot rule out the possibility of an extra proton. I recommend the authors should indicate that observation in the main text and include a reference to earlier work where that possibility has already been supported by direct comparisons to experimental data.

Response: Thank you for your additional comments and giving us a chance to explain. The two specific papers supporting W2=OH⁻ are invoked here mainly because two of

our co-authors (J. M. and D. P.) of the present study were also contributors therein, but there are many other publications by other groups suggesting or adopting $W2=OH^{-1-15}$ (see these references for examples). Your concerns are valid regarding that DFT calculations can sometimes struggle with accurately predicting spin states, especially in such complex and heavily coupled systems as the OEC, due to the effects including electron correlation, functional/basis dependence, spin contamination, protonation states and hydrogen-bonding network, etc. Thus, the situation is close to the limits of the method as such and one cannot expect DFT calculations under such circumstances to provide a definitive answer. Indeed, a comparative analysis of other protonation states would be meaningful but it is hardly conceivable to replicate the complete study using different protonation states in the current study. It is of course worthwhile to be investigated in a separate and complete study which specially focuses on the effects of the protonation states of W2, possibly as well including other ligands. However, that will be a major computational study on its own well beyond the scope of the current study. Here, we are just using the scenario that in our views (and also accepted by other groups) is suggested as highly consistent with experimental magnetic and EPR spectroscopic data in previous work. We believe that the exploration process represents a normal and sensible strategy for such a complex system with unsettled issues, in which certain assumptions are also made in other theoretical modelling work in this field. However, as you indicated, we acknowledge that the issue of the protonation state of $W2=H_2O/OH^{-}$ is not settled. For example, the calculations that attempted to reproduce other types of spectroscopies (FTIR and XAS respectively)^{16,17} favored a doubly protonated W2, and there are also other studies adopting $W2=H_2O$ in certain S states¹⁸⁻²⁷. We fully respect the scientific significance although it is different from our selected model in this study. Therefore, according to your suggestion, we have now explicitly stated the possibility of $W2=H_2O$ in the main text with three selected references (the first paragraph in the ‘Results’ section), and the corresponding content in Text S3 in the Supplementary information has also been adjusted with more relevant references, and the annotation of Fig. 4 has also been modified.

(2) Regarding W1 breaking the hydrogen bond with D61 to rotate into the position of W2, they do emphasize that this happens only within 1-2 ps in their MD. I guess Mn4 being a Mn³⁺ makes it possible to rotate W1, compared to Mn4 being Mn⁴⁺ with a quantum proton in between W1 and D61. However, it is unclear whether that rearrangement is induced by the specific preparation of the initial conditions. I recommend the authors should analyze how much of that rearrangement depends on the initial conditions of their simulations.

Response: Thank you for your additional suggestion. First of all, we feel it necessary to re-emphasize that the H-bond interaction between W1 and D61 is actually not broken, which was shown in the last response to your question but we might not have been clear enough. In that response we stated “In this process, the H-bond interaction between W1 and D61 is actually not broken because the cluster is expanded by the elongated Mn1-Mn4 distance (or by enlarged \angle Mn3-O4-Mn4) and the W1-D61 interaction is maintained (although W1 rotates some angle around Mn4) and this is another clear difference from the ‘pivot/carousel’ mechanism for S₂-S₃.” One can also consult Fig. 2 (Im2) on this point or water the uploaded ‘Movie’ for the dynamic images for the process. For the short timescale of the ligand rearrangement, you are correct that Mn4 being a Mn(III) is an important factor facilitating the W1/W2 rotation on Mn4, and another advantageous factor is the W3-W2 HB interaction. We had also already shown these in the last response: “We surmise the more facile ligand reorganization suggested here could be attributed to two factors that are absent in the ‘pivot/carousel’ mechanism for S₂-S₃: valence (III) of Mn4 and the W3-W2 HB interaction. The bond strength of ligands (H₂O) on Mn4(III) should be less than that of OH⁻ on Mn4(IV), so that ligand displacement around Mn4(III) can be easier. Besides, W2 is connected by the strong H-bond interaction to W3 which is bonded to Mn1, making W2 rotate some angle around Mn4 toward the cavity in Im1. W1 is also affected and moves closer to the para-position of O4 for a more stabilized geometry and thus create enough room to accommodate W7.

W2 protonation from W3 weakens the ‘structural *trans* effect’ of Mn4 and significantly facilitates W7 binding. These are supposed to account for the smooth and barrierless translocations of water ligands on Mn4(III) observed in the MD simulations.”

Regarding your question that whether the rearrangement is induced by the specific preparation of the initial conditions, it is certain that any theoretical modelling has to be started by a specific preparation of ‘initial conditions’, as long as they are well-founded in the current knowledge system and based on reasonable assumptions for unsolved issues. The original structure is fundamentally based on the room-temperature serial femtosecond crystallography of the S₃ state, and the corresponding S₄ state was optimized after removing one proton and one electron from Ox (assumed as a hydroxide in S₃). Then the Im0^{-O₂} structure was produced and optimized by removal of O5 and Ox. Thus, from S₃ to S₄ to Im0^{-O₂}, we have followed a normal and logical procedure for producing the coordinates of the Im0^{-O₂} intermediate. Please note that direct experimental observation for the Im0^{-O₂} state (as well as the S₄ and S₄-peroxide states) has not been realized so far, and thus in a computational study we have to construct a model based on comprehensive consideration of all available knowledge in this field. Since the present study is focused on how the cluster would evolve after O₂ release, the available ‘evidence’ or strong support for the existence of the Im0^{-O₂} state is mainly reflected in two aspects: 1) O5 and Ox as substrates for O-O bond formation, and 2) the complete removal of O₂ from the cluster for the subsequent entrance of a water molecule, which have been more extensively demonstrated in Supplementary Text S1 and Text S4, respectively. In this sense, our Im0^{-O₂} model, in terms of a cavity inside the Mn₄CaO₄ cluster, makes sense for the structural evolution during the S₄-S₀ transition after O₂ release.

It can be noted that the most relevant specific ‘initial conditions’ of our modelling mainly include: 1) O5 and Ox as substrates for O-O bond formation, 2) fully O₂ dissociative process of water insertion, and 3) the hydroxyl protonation state of W2. Therefore, we think your additional question is still related to our last response to your first comments, where we presented the theoretical basis in detail and we do not

elaborate on that issue here. For the points 1) and 2), we assume we have reached an agreement. For the point 3), we respect your reservation of 'W2=H₂O' although we favor 'W2=OH⁻' instead; we highlight that this remains an open question and we have made necessary revisions according to your suggestion, as shown above. With respect to the effect on the rearrangement, besides the Mn³⁺ nature of Mn4, the HB interaction of W3(H₂O)-W2(OH⁻) indeed makes a contribution. W2 is connected by the strong H-bond interaction to W3 which is bonded to Mn1, making W2 rotate to some degree around Mn4 toward the cavity in Im1. W1 is also affected and moves closer to the para-position of O4 for a more stabilized geometry and thus create enough room to accommodate W7. W2 protonation weakens the 'structural trans effect of Mn4 and significantly facilitates W7 binding, further promoting the 'pivot/carousel'-like ligand rearrangement. These effects are supposed to account for the smooth and barrierless translocations of water ligands on Mn4(III) observed in the MD simulations. These are observations and considerations for W2=OH⁻ in the 'initial condition', and it is uncertain if similar rearrangement would also take place when W2=H₂O. We surmise the W3(H₂O)-W2(H₂O) HB interaction would be somewhat weaker than in the present W3(H₂O)-W2(OH⁻) case, but this may not necessarily block the W3 deprotonation to W2 and ligand rearrangement on Mn4(III) because W2 may deprotonate to Asp61 (to lumen) before accepting a proton from W3; this may not be captured in a picosecond MD simulation and MEP calculations may be needed...As we have indicated above, the alternative situation of 'W2=H₂O' would be worth trying in a separate complete study. However, at present, we believe the situation of 'W2=OH⁻' represents a meaningful and credible investigation for water insertion (the recent breakthrough by Greife et al.¹ also adopts such a protonation scheme).

Another variable factor, not referring to the 'initial conditions' itself but still being a relevant one, is the model/methodology adopted, i.e. QM/MD on a sufficiently large cluster model versus QM/MM MD including a much larger protein environment (but possibly a smaller QM region). The latter approach would represent a totally different methodology, and it is uncertain if it is necessary to include the distal residues to explore

the Mn cluster evolution as in the case of the present study. It would also be worthwhile to explore in a full but separate study. but the QM/MD simulations here have taken into account all the necessary H-bond interactions of the key groups: W3 is H-bonded to W5 and HOH511 which are further H-bonded to W6 which is further H-bonded to Asn181; W7 is H-bonded to O4, W8, the guanidine group of Arg357, amide oxygen of Asp170 and alcoholic hydroxyl of the side chain of Asp170; W8 is H-bonded to HOH574, amide oxygens of Asp170 and Asn87. Since all these H-bonded joint groups are explicitly included in the model by QM description, their influence on the movement of W3, W7, and W8 should have been embodied in the MD simulations. Thus, it can be noted that even for the most distal waters such as W6 and W8, the H-bond interactions with other surrounding waters or protein residues are also explicitly included. The more distant (approximately >8 Å away from the cluster) water molecules and protein residues that are not included in the model do not have direct H-bond interactions to any of W3, W7 and W8; they may produce very minor electrostatic and van der Waals interactions that we assume as insignificant and have little impact on the movement of W3, W7, and W8. Therefore we believe our model constitutions for the starting structure are adequate to characterize the chemical environment responsible for the proposed rearrangement.

Apart from the above points, we do not see how the ‘initial conditions’ could substantially vary to other justifiable models that originate from compelling experimental/theoretical suggestions and that would significantly change the subsequent evolution of the proposed rearrangement. As the spin orientation on Mn2(IV) does not play a significant role in the process, the almost identical simulation observations for the two spin states actually represent a mutual testification eliminating the possibility of randomness of the simulation results, so that the common phenomenon captured for the Mn4 ligand rearrangement is reproducible and reliable. In the main text page 8, one paragraph is presented to illustrate the molecular principles behind the water insertion pathway and the accompanying ligand rearrangement on Mn4. We believe the special rearrangement is a common/reasonable event rather than

a rare occurrence or even an artifact created by specific ‘initial conditions’. The model is based on a well-defined starting structure and there seems not much freedom to vary the ‘initial conditions’ in the present study. The above arguments are briefly reflected in the main text and in detail in Text S1, Text S3, Text S4, Text S6, Text S7, Text S10, Text S14 and Text 17 in the Supplementary information. We hope these explanations can fully address your concerns.

(3) The authors have estimated how much energy is required to detach W2. However, they have not estimated the error of that estimated energy. I recommend they should provide an estimate of the error of their calculation.

Response: Thank you for your additional question. We think the case here actually belongs to a rather general consideration with respect to the application of DFT methodology to chemical reactions in transition metal chemistry that has been well-addressed in many papers²⁸⁻³⁵. As shown in Fig. 4, the energetic barriers for W2 decoupling are calculated to be 4~5 kcal/mol and endothermic by 2~3 kcal/mol by a truncated DFT model using MEP-IRC calculations including 200+ atoms. According to previous studies^{36,37}, a QM model with such a size is adequate to yield accurate reaction energetics and spectroscopic properties for the OEC system. It is acknowledged in many studies/benchmarks that DFT computations can give quite reliable results with errors normally within the range of approximately 1-3 kcal/mol as compared to more advanced (but much more expensive) methods. As we note, there would be possibly some error here, and we have explicitly included a discussion on potential errors in Text S10 in the Supplementary information. Given that no sampling of different trajectories is performed multiple times on one spin state, we have only a single value for one spin state without sampling of many trajectories. Therefore we cannot make a statistical estimation in this particular case because there is no concept of standard deviation around a mean value, and hence there can be no uncertainty quantification in the conventional sense, or providing the ‘error bars’ associated with

the reported computed value. Nevertheless, we have two different spin state models extracted from two different simulation trajectories which obtained very similar energetics for W2 decoupling from, thus it is expected that multiple sampling would not change the situation and the error in the estimated energy would not be significantly high to affect the conclusion. Practically, a more meaningful assessment on the results can be performed by sensitivity tests using different functionals, with the basic motivation that different functionals represent different approximations that could affect the quantitative results obtained. This has been shown in Tables S5-S6 and Fig. S11 in the Supplementary information, together with the sensitivity tests for the isomerism. For W2 decoupling as you concern, five more GD3BJ-parameterized DFT functionals yield 3.4~4.9 kcal/mol barriers and endothermic by 2.5~4.0 kcal/mol, which means merely 1~2 kcal/mol deviation. For the isomerism, even changing the source of the model by employing 6DHP of the original S_0 state yields very similar energetics (deviation within 1 kcal/mol), which indicates very small errors which would not affect the qualitative conclusion and also reliability of the model used. Besides, the obtained barrier height for W2 detaching from Mn4(III) is quite similar to a recent report on a biomimetic polyoxometalate water oxidation catalyst³⁸, supporting the rationality of the obtained energetics. We have modified the relevant part on page 12 of the main text and also added some discussion in Text S10 of the Supplementary information. We hope these can address your concerns.

Reviewer #3 (Remarks to the Author):

Authors have convincingly addressed all the critical points I had indicated. Therefore, I suggest the publication of the manuscript in its current form in the Nature Communications journal.

I only make one note: in their response to the referees (see, for example, referee 1), the authors state that the work by Guidoni and coworkers is based on QM/MM MD Car-

Parrinello. In reality, the works in question were carried out based on QM/MM MD Born-Oppenheimer as implemented in the CP2K package.

Response: Thank you again for your nice review and the correction! We have corrected the statement for all the places in the Supplementary information.

Reviewer #4 (Remarks to the Author):

The revised version is acceptable to me.

Response: Thank you again for your nice review!

References

1. Greife, P., *et al.* The electron–proton bottleneck of photosynthetic oxygen evolution. *Nature* **617**, 623–628 (2023).
2. Capone, M., Narzi, D. & Guidoni, L. Mechanism of oxygen evolution and Mn₄CaO₅ cluster restoration in the natural water-oxidizing catalyst. *Biochemistry* **60**, 2341–2348 (2021).
3. Siegbahn, P.E.M. Water oxidation mechanism in photosystem II, including oxidations, proton release pathways, O–O bond formation and O₂ release. *Biochim. Biophys. Acta, Bioenerg.* **1827**, 1003-1019 (2013).
4. Shevela, D., Kern, J.F., Govindjee, G. & Messinger, J. Solar energy conversion by photosystem II: principles and structures. *Photosynth. Res.* **156**, 279–307 (2023).
5. Song, X. & Wang, B. O–O bond formation and oxygen release in photosystem II are enhanced by spin-exchange and synergetic coordination interactions. *J. Chem. Theory Comput.* **19**, 2684–2696 (2023).
6. Shoji, M., Isobe, H., Shigeta, Y., Nakajima, T. & Yamaguchi, K. Concerted mechanism of water insertion and O₂ release during the S₄ to S₀ transition of the oxygen-evolving complex in photosystem II. *J. Phys. Chem. B* **122**, 6491-6502 (2018).
7. Bovi, D., Narzi, D. & Guidoni, L. The S₂ state of the oxygen-evolving complex of photosystem II explored by QM/MM dynamics: spin surfaces and metastable states suggest a reaction path towards the S₃ state. *Angew. Chem. Int. Ed.* **52**, 1-6 (2013).
8. Robertazzi, A., Galstyan, A. & Knapp, E.W. PSII manganese cluster: protonation of W2, O5, O4 and His337 in the S₁ state explored by combined quantum chemical and electrostatic energy computations. *Biochim. Biophys. Acta, Bioenerg.* **1837**, 1316-1321 (2014).
9. Corry, T.A. & O'Malley, P.J. Molecular identification of a high-spin deprotonated intermediate during the S₂ to S₃ transition of nature's water-oxidizing complex. *J. Am. Chem. Soc.* **142**, 10240-10243 (2020).

10. Galstyan, A., Robertazzi, A. & Knapp, E.W. Oxygen-evolving Mn cluster in photosystem II: the protonation pattern and oxidation state in the high-resolution crystal structure. *J. Am. Chem. Soc.* **134**, 7442-7449 (2012).
11. Pushkar, Y., Ravari, A.K., Jensen, S.C. & Palenik, M. Early binding of substrate oxygen is responsible for a spectroscopically distinct S₂ state in photosystem II. *J. Phys. Chem. Lett.* **10**, 5284-5291 (2019).
12. Isobe, H., Shoji, M., Shen, J.-R. & Yamaguchi, K. Chemical equilibrium models for the S₃ state of the oxygen-evolving complex of photosystem II. *Inorg. Chem.* **55**, 502-511 (2016).
13. Shoji, M., Isobe, H. & Yamaguchi, K. QM/MM study of the S₂ to S₃ transition reaction in the oxygen-evolving complex of photosystem II. *Chem. Phys. Lett.* **636**, 172-179 (2015).
14. Huo, P.-Y., Jiang, W.-Z., Yang, R.-Y. & Zhang, X.-R. Dynamic evolution of S₀-S₃ at the oxygen-evolving complex with spin markers under photoelectric polarization. *Phys. Rev. Appl.* **21**, 024024 (2024).
15. Drosou, M. & Pantazis, D.A. Comprehensive evaluation of models for ammonia binding to the oxygen evolving complex of photosystem II. *J. Phys. Chem. B* **128**, 1333-1349 (2024).
16. Yamamoto, M., Nakamura, S. & Noguchi, T. Protonation structure of the photosynthetic water oxidizing complex in the S₀ state as revealed by normal mode analysis using quantum mechanics/molecular mechanics calculations. *Phys. Chem. Chem. Phys.* **22**, 24213-24225 (2020).
17. Chrysinia, M., *et al.* Nature of S-states in the oxygen-evolving complex resolved by high-energy resolution fluorescence detected X-ray absorption spectroscopy. *J. Am. Chem. Soc.* **145**, 25579–25594 (2023).
18. Yang, K.R., Lakshmi, K.V., Brudvig, G.W. & Batista, V.S. Is deprotonation of the oxygen-evolving complex of photosystem II during the S₁ → S₂ transition suppressed by proton quantum delocalization? *J. Am. Chem. Soc.* **143**, 8324-8332 (2021).
19. Rummel, F. & O'Malley, P.J. How nature makes O₂: an electronic level mechanism for water oxidation in photosynthesis. *J. Phys. Chem. B* **126**, 8214–8221 (2022).
20. Isobe, H., *et al.* Theoretical illumination of water-inserted structures of the CaMn₄O₅ cluster in the S₂ and S₃ states of oxygen-evolving complex of photosystem II: full geometry optimizations by B3LYP hybrid density functional. *Dalton Trans.* **41**, 13727-13740 (2012).
21. Nakamura, S. & Noguchi, T. Quantum mechanics/molecular mechanics simulation of the ligand vibrations of the water-oxidizing Mn₄CaO₅ cluster in photosystem II. *Proc. Natl. Acad. Sci. U.S.A.* **113**, 12727-12732 (2016).
22. Yamaguchi, K., *et al.* Geometric, electronic and spin structures of the CaMn₄O₅ catalyst for water oxidation in oxygen-evolving photosystem II. Interplay between experiments and theoretical computations. *Coord. Chem. Rev.* **471**, 214742 (2022).
23. Yamaguchi, K., *et al.* Theoretical elucidation of the structure, bonding, and reactivity of the CaMn₄O_x clusters in the whole Kok cycle for water oxidation embedded in the oxygen evolving center of photosystem II. New molecular and quantum insights into the mechanism of the O–O bond formation. *Photosynth. Res.* (2023).
24. Pal, R., *et al.* S₀-State model of the oxygen-evolving complex of photosystem II. *Biochemistry* **52**, 7703-7706 (2013).
25. Saito, K., Nakao, S. & Ishikita, H. Identification of the protonation and oxidation states of

- the oxygen-evolving complex in the low-dose X-ray crystal structure of photosystem II. *Front. Plant Sci.* **14**(2023).
26. Kawashima, K., Takaoka, T., Kimura, H., Saito, K. & Ishikita, H. O₂ evolution and recovery of the water-oxidizing enzyme. *Nat. Commun.* **9**, 1247 (2018).
 27. Isobe, H., *et al.* Generalized approximate spin projection calculations of effective exchange integrals of the CaMn₄O₅ cluster in the S₁ and S₃ states of the oxygen evolving complex of photosystem II. *Phys. Chem. Chem. Phys.* **16**, 11911-11923 (2014).
 28. Cramer, C.J. & Truhlar, D.G. Density functional theory for transition metals and transition metal chemistry. *Phys. Chem. Chem. Phys.* **11**, 10757-10816 (2009).
 29. Siegbahn, P.E.M. The performance of hybrid DFT for mechanisms involving transition metal complexes in enzymes. *J. Biol. Inorg. Chem.* **11**, 695–701 (2006).
 30. Siegbahn, P.E.M. & Blomberg, M.R.A. Transition-metal systems in biochemistry studied by high-accuracy quantum chemical methods. *Chem. Rev.* **100**, 421-437 (2000).
 31. Claeysens, F., *et al.* High-accuracy computation of reaction barriers in enzymes. *Angew. Chem. Int. Ed.* **45**, 6856 -6859 (2006).
 32. Harvey, J.N., Poli, R. & Smith, K.M. Understanding the reactivity of transition metal complexes involving multiple spin states. *Coord. Chem. Rev.* **238-239**, 347-361 (2003).
 33. Neese, F. Prediction of molecular properties and molecular spectroscopy with density functional theory: From fundamental theory to exchange-coupling. *Coord. Chem. Rev.* **253**, 526-563 (2009).
 34. Cheong, P.H.-Y., Legault, C.Y., Um, J.M., Çelebi-Ölçüm, N. & Houk, K.N. Quantum mechanical investigations of organocatalysis: mechanisms, reactivities, and selectivities. *Chem. Rev.* **111**, 5042-5137 (2011).
 35. Cheng, G.-J., Zhang, X., Chung, L.W., Xu, L. & Wu, Y.-D. Computational organic chemistry: bridging theory and experiment in establishing the mechanisms of chemical reactions. *J. Am. Chem. Soc.* **137**, 1706-1725 (2015).
 36. Retegan, M., Neese, F. & Pantazis, D.A. Convergence of QM/MM and cluster models for the spectroscopic properties of the oxygen-evolving complex in photosystem II. *J. Chem. Theory Comput.* **9**, 3832-3842 (2013).
 37. Blomberg, M.R.A., Borowski, T., Himo, F., Liao, R.-Z. & Siegbahn, P.E.M. Quantum chemical studies of mechanisms for metalloenzymes. *Chem. Rev.* **114**, 3601-3658 (2014).
 38. Schwiedrzik, L., Rajkovic, T. & González, L. Regeneration and degradation in a biomimetic polyoxometalate water oxidation catalyst. *ACS Catal.* **13**, 3007-3019 (2023).

REVIEWERS' COMMENTS

Reviewer #2 (Remarks to the Author):

The authors have addressed my earlier concerns and the manuscript can now be recommended for publication.

Reviewer #2 (Remarks to the Author):

The authors have addressed my earlier concerns and the manuscript can now be recommended for publication.

Response: Thank you again for your nice review!